# Personal, social, and natural co-exposure pattern and plasma proteins in cardiometabolic diseases

Xuewei Tang[1,6], Huan Xu[1,6], Gonghua Wu[1,6], Shaokun Yang[1], Yonghua Ye[1], Yan An[2], Yilin Jiang[3], Zilong Wang[4,5], Sujun Chen ®[1], Juying Zhang[1] ✉, Xiong Xiao ®[1] ✉, Bing Guo[1] ✉ & Xing Zhao ®[1] ✉

Multiple environmental exposures elevate the risk of cardiometabolic diseases (CMDs), yet the impact of combined co-exposures, and their protein-mediated mechanisms remain underexplored. Among 366,261 UK Biobank participants, this study identified five distinct exposure patterns through clustering analysis: Air and Noise Pollution, Social Deprivation, Blue and Green Space, Health Behaviors and Reference. Notably, the Social Deprivation pattern showed the highest CMDs risk, with hazard ratios ranging from 1.15 to 1.40. Inflammatory and immune proteins were associated with all co-exposure patterns. The Air and Noise Pollution and Social Deprivation patterns demonstrated that the common mediating proteins included CDCP1, CXCL17, FGF21, GDF15 and IGFBP4. CXCL13, CA6, SDC1, and PTN mediated Healthy Behaviors, while MMP12 mediated Blue and Green Space pattern. This study offers etiological insight into the comprehensive human environment, draws attention to the population experiencing health disadvantages as a result of co-exposure to social stressors, and elucidates protein-mediated mechanisms linking environmental exposures to disease.

With rapid industrialization and urbanization, cardiometabolic diseases (CMDs) have increased in prevalence for decades and emerged as the major public health challenges in high-income countries[1-4]. CMDs are marked by systemic interdependencies and share multiple common risk factors. Collectively, environmental components such as air pollution, noise, green spaces, lifestyle factors, and socioeconomic status have all been identified as common risk factors for CMDs[5-8].

Most research has primarily targeted specific individual exposures, ignoring the complexity of mixed exposures across natural, social, and personal domains[9,10]. Such isolated approaches risk underestimating or distorting the true health impacts. In real-world co-exposure environments, multiple exposures often interact synergistically or cumulatively, modifying their health impacts when considered alone or in combination[11-15]. This has led to the emergence of the study of exposome, a concept that covers cumulative exposures to physical, chemical, biological and psychosocial factors over a lifetime[16,17]. Moreover, widespread exposures usually exhibit a notable colocalization. For example, the distribution of light pollution, air pollution, and traffic noise levels exhibit significant geographical overlap[18]. Furthermore, individuals with low socioeconomic status are more prone to engage in detrimental lifestyle behaviors[19]. Therefore, given the broad range of exposures, capturing multidimensional co-exposure

[1]West China School of Public Health and West China Fourth Hospital, Sichuan University, Chengdu, China. [2]School of Life Sciences, Sun Yat-sen University, Guangzhou, China. [3]Frontier and International Psychiatry, Graduate School of Medicine, Kyoto University, Kyoto, Japan. [4]Department of Chemical Pathology, Prince of Wales Hospital, The Chinese University of Hong Kong, Shatin, New Territories, Hong Kong Special Administrative Region, China. [5]Li Ka Shing Institute of Health Sciences, The Chinese University of Hong Kong, Shatin, New Territories, Hong Kong Special Administrative Region, China. [6]These authors contributed equally: Xuewei Tang, Huan Xu, Gonghua Wu. ✉e-mail: juying109@163.com; xiaoxiong.scu@scu.edu.cn; guobing0111@foxmail.com; xingzhao@scu.edu.cn

patterns is essential for understanding the complex relationships between environmental exposures and cardiometabolic health. It facilitates a more comprehensive exploration of the joint effects of multiple exposures, the revelation of the environmental etiologies of diseases, and the provision of a basis for the sorting of optimal public health interventions and targeted populations.

Plasma proteins, in addition to serving as therapeutic and predictive disease biomolecules[20,21], can provide insights into the intrinsic mechanisms associated with environmental exposures to human health conditions and cardiometabolic diseases[22,23]. According to the findings of previous studies, which explored its function at the molecular level in the pathogenesis of various environmental exposures[24], the question remains unanswered whether it can similarly reflect the physiological responses to co-exposure patterns and whether it can reveal key biological pathways and mechanisms of the effect on CMDs.

This work aims to identify the co-exposure patterns and their specific effects on CMDs risk in UK Biobank dataset. Additionally, we are committed to unraveling the proteomics pathways that connect environmental exposures with the etiology of CMDs (Fig. 1).

## Results

### Population characteristics and individual exposure contributions to CMDs

A total of 366,261 participants were included in the cluster analysis of exposure patterns, of whom 46.4% were male and the vast majority were white (94.9%), with an average age of 56.43 years. In terms of exposure, the participants' average exposure to nitrogen oxides ($NO_x$), nitrogen dioxide ($NO_2$), particulate matter with a diameter of 10 micrometers or less ($PM_{10}$) and particulate matter with a diameter of 2.5 micrometers or less ($PM_{2.5}$) was 43.65, 26.48, 16.19 and 9.96 μg/m³, respectively. The 24-hour average noise level was 56.02 decibels, and the total water equivalent was low (0.88). More than half had never smoked (54.8%). After further exclusions, the data sets for cardiometabolic outcomes—heart disease, cerebrovascular disease, renal disease, diabetes, and death— included 336,778, 353,264, 353,892, 341,529, and 358,872 participants, respectively. The cumulative incidence for these diseases in each analysis set were 12.8%, 4.2%, 7.0%, 4.1%, and 6.7%, respectively, while the distribution of other variables remained similar to that in the cluster analysis data set (Supplementary Table 1).

As shown in Fig. 2a, we observed different magnitudes of effect for different exposures on the same CMD. Of particular note was the association of current smoking with death, which showed a significantly increased risk compared to never smoking (hazard ratio, HR = 2.76, 95% CI: 2.66- 2.86). Conversely, exposure to green space and adherence to healthy lifestyle factors—such as 7 h of sleep, a balanced diet and adequate physical activity—showed protective associations against the risk of CMDs. In the multi-exposure models (Supplementary Fig. 1), the effects of several exposures differed from those of the single-exposure models. To account for the collinearity, interactions and potential non-linear relationships, we further applied the XGBoost model. In Supplementary Fig. 2, each CMD model achieved a C-index above 0.7, confirming model robustness. For heart disease, the top three exposures were current smoking, anxiety or(and) depression, and former smoking (absolute SHapley Additive exPlanations (SHAP) values: 0.070, 0.068, and 0.049, respectively). The Index of Multiple Deprivation (IMD) ranked among the top three exposure variables for cerebrovascular disease, diabetes, and renal disease (Fig. 2b).

### Clustering of Co-exposure patterns and impact on CMD Risks

As the Fig. 3a and Supplementary Table 2 shown, five distinct subgroups were identified by applying the K-prototypes clustering method to co-exposure patterns. The Air and Noise Pollution group, representing only 5.78% of the population for clustering analysis, had elevated average levels of $NO_x$, $NO_2$, $PM_{10}$, $PM_{2.5}$, and 24-hour noise (79.05 μg/m³, 39.95 μg/m³,18.96 μg/m³, 11.52 μg/m³, and 68.08 dB, respectively), while averages for other variables in this group were notably low. The Social Deprivation group, making up 20.52% of the

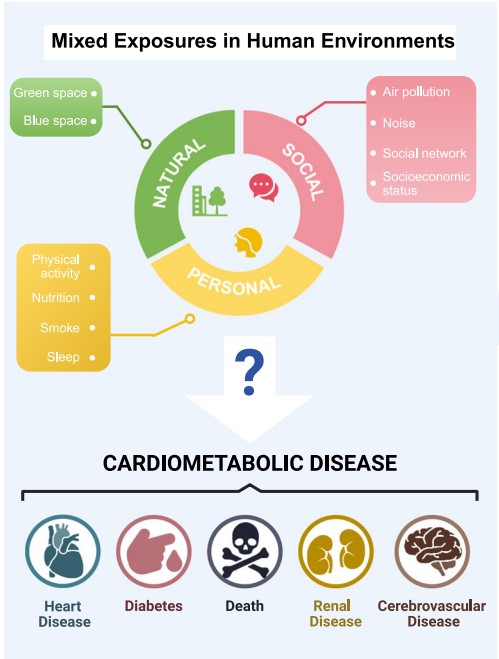

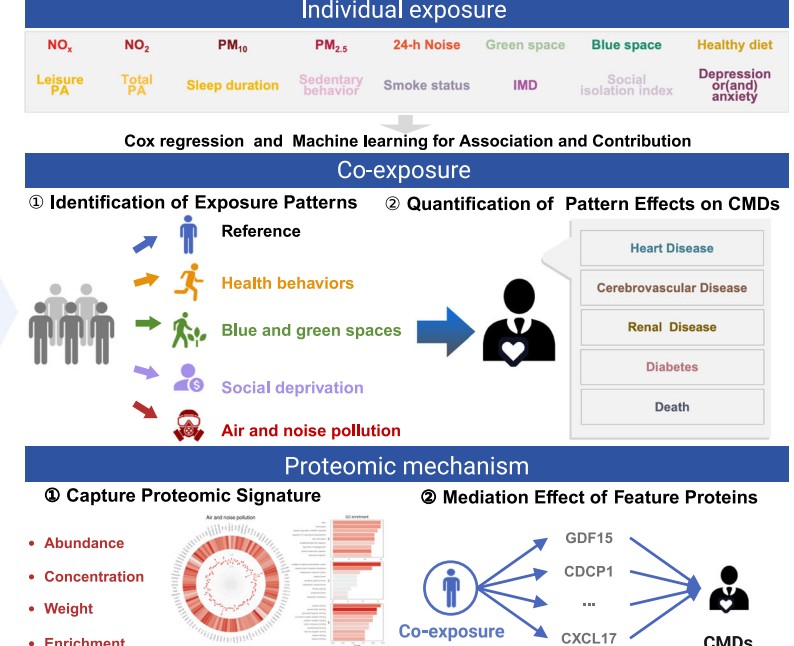

**Fig. 1 | Overview of study design.** We assessed the contributions of individual, social, and natural environmental exposures to cardiometabolic diseases using Cox regression and XGBoost model. Participants were then classified into 5 distinct co-exposure pattern subgroups via K-prototypes clustering, and the effects of these patterns were estimated using Cox regression. Leveraging high-dimensional proteomic data, we analyzed 2,911 proteins to identify proteomic signatures associated with each co-exposure pattern using LASSO regression. Additionally, mediation analysis was applied to pinpoint key proteins that mediate the relationship between these exposure patterns and CMD outcomes. *CMD* cardiometabolic disease; *PM* particulate matter; *IMD* Index of Multiple Deprivation; *PA* physical activity. Created in BioRender. Xw, T. (https://BioRender.com/ebkarjr).

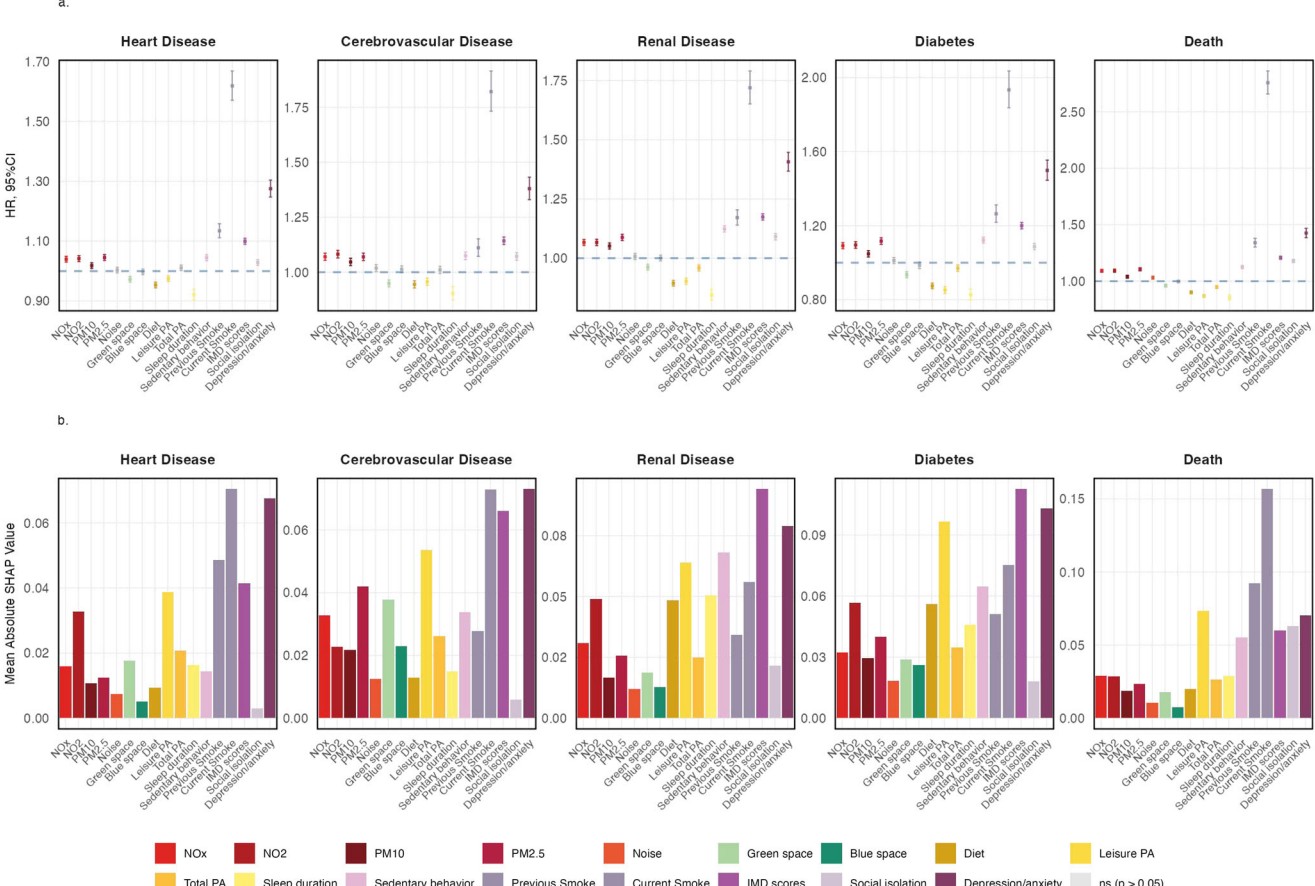

**Fig. 2 | Individual exposure effect and contribution on cardiometabolic diseases. a** Hazard ratios (HRs) were estimated using single-exposure multivariable Cox proportional hazards regression models, adjusted for age, sex, ethnicity, alcohol consumption status, and BMI. Each exposure was standardized prior to modeling. Circular markers represent HR point estimates for each association between an individual exposure and an outcome. Vertical error bars represent 95% confidence intervals (CIs). The horizontal dashed line denotes the null value of HR = 1. Colors denote the type of exposure, with gray denoting exposures with an adjusted *p* value > 0.05. Statistical significance was assessed using two-sided Wald tests with Bonferroni correction for multiple comparisons. **b** Relative importance of each standardized exposure variable for CMDs risk, based on absolute SHapley Additive exPlanations (SHAP) values calculated using XGBoost models. BMI, body mass index; PM, particulate matter; IMD, Index of Multiple Deprivation; PA, physical activity. The sample sizes (independent UK Biobank participants) for the cardiometabolic outcomes were *n* = 336,778 for heart disease, *n* = 353,264 for cerebrovascular disease, *n* = 353,892 for renal disease, *n* = 341,529 for diabetes, and *n* = 358,872 for death. Source data are provided.

population, was characterized by high IMD scores (32.75), a high social isolation index (0.96), extensive sedentary time (4.33 h/day), and high prevelence of anxiety/depression (31.19%) and current smoking (16.20%). The Blue and Green Space group, encompassing 23.14% of participants, exhibited high averages in green (66.53 percent) and blue (1.63 percent) spaces. In the Health Behaviors group, representing 9.42% of the population, individuals showed high dietary scores, leisure and total physical activity (2.71, 3802.18 MET, and 8142.41 MET, respectively), with over half (53.60%) achieving optimal sleep duration. The remaining 23.14% of individuals were classified as the Reference group, characterized by exposure levels that were generally around the average across all subgroups.

In Fig. 3b, we examined the association between these clustered co-exposure patterns and CMDs. Both the Air and Noise Pollution and Social Deprivation groups demonstrated adverse effects across CMDs, though with varying degrees of risk. For instance, individuals in the Air and Noise Pollution group had an HR of 1.06 for heart disease, indicating a 6% higher risk compared to the reference group. Elevated HRs were also observed for cerebrovascular disease, renal disease, diabetes, and death, ranging from 1.06 to 1.19. Meanwhile, the Social Deprivation group showed a more pronounced risk for all CMDs, with HRs ranging from 1.15 to 1.40. Conversely, exposure to Blue and Green Space had a protective association, particularly for diabetes (HR =

0.89, 95% CI: 0.85–0.93). Additionally, the Health Behaviors group showed the greatest benefits for death, diabetes, and renal disease, with HRs of 0.88 (0.84–0.93), 0.86 (0.81–0.92), and 0.89 (0.85–0.94), respectively.

**Proteomics analysis**

In the proteomic analysis, 37,687 participants were categorized into five distinct exposure subgroups, enabling assessment of associations between 2911 proteins and CMDs across these groups (Supplementary Data 1, Supplementary Data 2, Supplementary Fig. 3). Following *P* value adjustment, the Health Behaviors group exhibited the fewest death-associated proteins, with only 15 identified, while the Reference group had the most renal disease-associated proteins, totaling 741. We combined significant proteins from all diseases to obtain a total of 454 proteins to provide the basis for subsequent LASSO regression analysis. Relative to the Reference group, the distinct proteins identified in the Air and Noise Pollution, Social Deprivation, Blue and Green Space, and Health Behaviors patterns numbered 105, 168, 149, and 159, respectively (Supplementary Data 3). Figure 4 displays these proteins' signature weights in specific exposure pattern group. Enrichment analysis of biological processes showed that in the Air and Noise Pollution group, top processes included taxis, positive MAPK cascade regulation, vasculature development regulation, and cell chemotaxis.

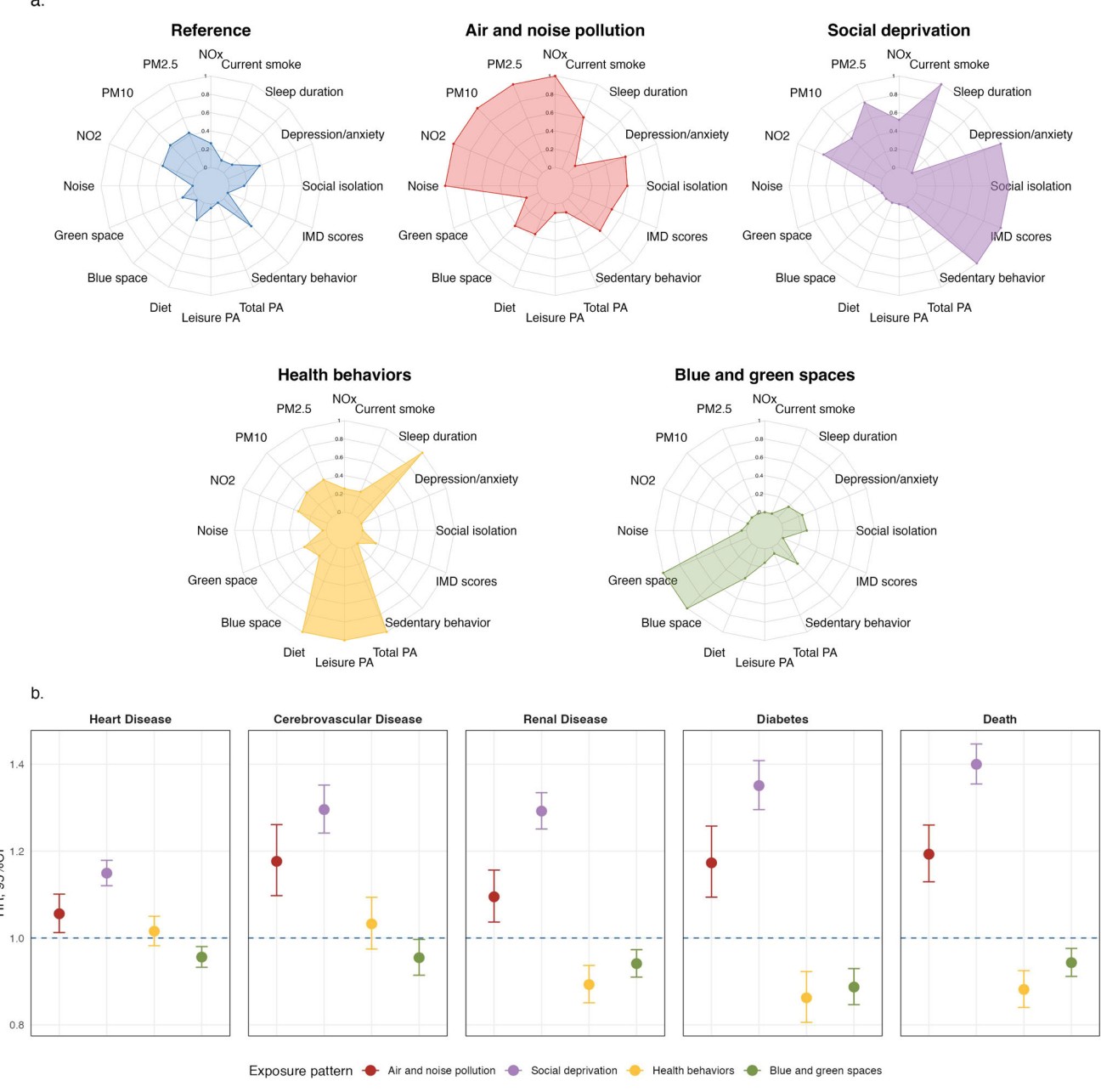

**Fig. 3 | Co-exposure pattern Characteristics and effects on Cardiometabolic Diseases(CMDs). a** Radar plot illustrating the characteristics of clusters identified through K-prototypes clustering. Each axis represents the average value of a distinct exposure variable within each pattern, with values scaled from 0 to 1 based on each exposure's minimum and maximum values. **b** Associations between co-exposure patterns and CMDs estimated using multivariable Cox proportional hazards regression, adjusted for age, sex, ethnicity, alcohol consumption status, and BMI. Circular markers represent hazard ratio (HR) point estimates, with vertical error bars denoting 95% confidence intervals (CIs). The horizontal dashed line indicates the null value of HR = 1, with the Reference pattern serving as the control group. Colors denote exposure patterns. Statistical significance was assessed using two-sided Wald tests (Exact *P* values are available in the corresponding Source Data file). The sample sizes (independent UK Biobank participants) for the cardiometabolic outcomes were *n* = 336,778 for heart disease, *n* = 353,264 for cerebrovascular disease, *n* = 353,892 for renal disease, *n* = 341,529 for diabetes, and *n* = 358,872 for death. *BMI* body mass index; *PM* particulate matter; *IMD* Index of Multiple Deprivation; *PA* physical activity. Source data are provided.

Notably, leukocyte migration ranked highly in the Social Deprivation group, while ameboid-type cell migration was prominent in the Blue and Green Space group. Regulation of vascular development and axis formation ranked first and third, respectively, in the Health Behaviors group. Additional enrichment result of GO Molecular Function (MF) and Cellular Component (CC) and KEGG are also shown in Supplementary Table 3 and Supplementary Table 4. The effect of proteomic signatures on CMDs demonstrated alignment with the correlations observed between co-exposure patterns and CMDs risk

(Supplementary Fig. 4), particularly through consistent risk elevations seen in the Air and Noise Pollution and Social Deprivation patterns. The observed alignment was attenuated in renal disease analyses, where the protective effect of Blue and Green Spaces differed from that of the proteomic signature.

In Fig. 5 and Supplementary Data 4, the feature proteins with substantial mediating effects are presented (adjusted *P* < 0.05 and proportion mediated stable), revealing key mediators between each exposure characteristic and CMDs. Proteins such as CDCP1, CXCL17,

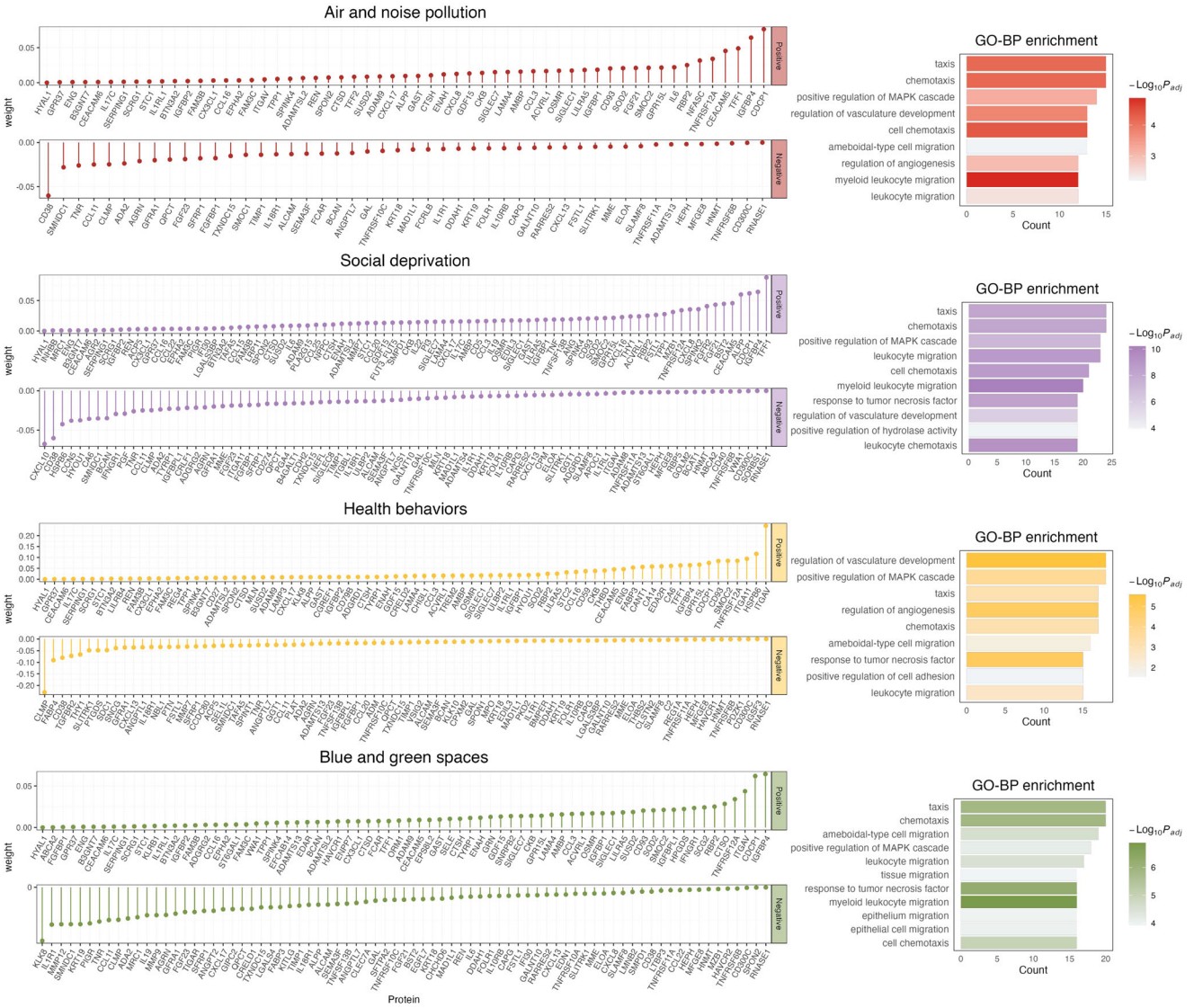

**Fig. 4 | Proteomic signature of co-exposure pattern.** Left: Feature proteins for each exposure pattern and their corresponding weights were identified using multinomial LASSO regression. This method is based on penalized maximum likelihood estimation and does not provide traditional null-hypothesis test statistics. The regularization parameter was selected via 10-fold cross-validation. Reported weights represent relative coefficients after shrinkage, with non-zero coefficients indicating proteins retained by the model. Right: Gene Ontology (GO) enrichment analysis was performed on the feature proteins to identify the associated biological processes (BP). Enrichment was assessed using a one-sided Fisher's exact test with Bonferroni correction for multiple testing. Only pathways with adjusted $p$ values < 0.05 are shown. Exact adjusted $p$ values are provided in Supplementary Table 3, and full protein names are listed in Supplementary Data 1. The sample sizes (independent UK Biobank participants) for the cardiometabolic outcomes were $n = 35,056$ for heart disease, $n = 37,025$ for cerebrovascular disease, $n = 36,963$ for renal disease, $n = 35,754$ for diabetes, and $n = 37,687$ for death.

FGF21, GDF15, and IGFBP4 commonly mediated effects in both the Air and Noise Pollution and Social Deprivation groups, while GDF15 was notably linked to heart disease across these patterns. ITGAV, HPGDS, ADGRG2, and MMP12 were mediators between Blue and Green Space pattern and death. Additionally, CXCL13, CA6, SDC1, and PTN demonstrated mediating effects in the association between Health Behaviors and outcomes such as diabetes, renal disease, and death. We provide a brief summary of the exposure patterns corresponding to the top five proteins in Supplementary Fig. 5.

### Sensitivity Analysis

First, sensitivity analyses using Cox proportional hazards models with categorized age and BMI demonstrated consistent associations between individual exposures and CMDs compared to our primary findings (Supplementary Table 5). Second, after changing the XGBoost machine learning parameter settings, the C-index for each disease was stable which indicated the robustness of analysis of individual exposure contributions (Supplementary Table 6). Third, the relationship between exposure patterns and diseases within individuals who had no pre-existing CMDs at baseline were largely consistent with relationship in whole population: Air and Noise Pollution and Social Deprivation had significant adverse effects on CMDs, while Health Behaviors and Blue and Green Space exhibited protective effects (Supplementary Table 7). The relationships between protein signatures and diseases in the subgroups also remained consistent (Supplementary Table 8). Fourth, to avoid reverse causation between proteins and diseases, we excluded participants with short-term onset of CMDs after the baseline survey. The directions of the associations between protein markers and diseases remained unchanged (Supplementary Table 9). Fifth, we adjusted for renal function in our analysis of protein characteristics and disease and found that the association between Blue and Green Space and renal disease was no longer statistically significant (p > 0.05)

 

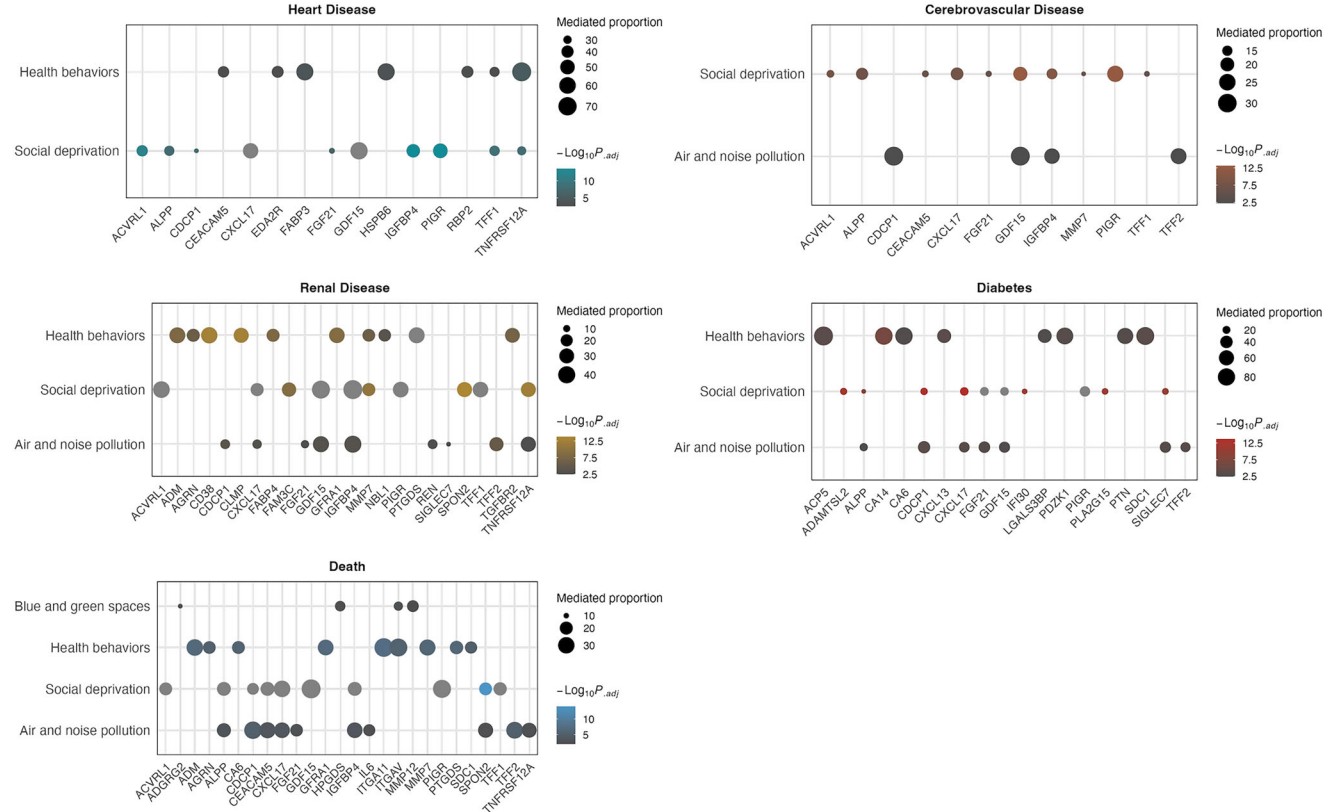

**Fig. 5 | Key Plasma Proteins Mediating the Effect of Co-Exposure Patterns on Cardiometabolic Diseases(CMDs).** Mediation proportions were estimated using a regression-based mediation approach with a Cox outcome model and a linear mediator model. *P* values for the natural indirect effects were calculated using two-sided Z-tests based on the delta method and adjusted for multiple comparisons using the Bonferroni method. After correction and exclusion of inconsistent mediation, only the top ten proteins with the highest mediated proportions and statistically significant adjusted *p* values (*p* < 0.05) for each exposure pattern–outcome pair are shown. Point size represents the magnitude of the mediation proportion, and color indicates the adjusted *p* value. Exact adjusted *p* values are provided in Supplementary Data 4, and protein names are provided in Supplementary Data 1. The sample sizes (independent UK Biobank participants) for the cardiometabolic outcomes were *n* = 35,056 for heart disease, *n* = 37,025 for cerebrovascular disease, *n* = 36,963 for renal disease, *n* = 35,754 for diabetes, and *n* = 37,687 for death.

(Supplementary Table 10). Sixth, mediation analyses were performed on all proteins. With the exception of ITGAV, HPGDS, and ADGRG2 for Blue and Green space, the Air and Noise pollution, Social Deprivation, and Health Behavior key proteins still remained significant mediating effects (Supplementary Data 5).

## Discussion

This study uncovers the complex cardiometabolic impacts of multi-faceted exposures across natural, social, and personal environments, by dissecting individual exposure contributions, characterizing co-exposure patterns, and probing potential protein mechanism. Our findings reveal that exposures exhibit co-localization, with distinct clustering patterns exerting varying influences on different diseases. By analyzing protein signature across different exposure patterns, we found that immune and inflammatory response pathways played a key role in all exposure patterns, especially in Social Deprivation and Air and Noise pollution patterns. Furthermore, our mediation analysis highlighted the mediating roles of several proteins in the relationship between exposure patterns and diseases. These findings not only provide insights into the multi-dimensional impact of environmental and social exposures on CMDs, but also unravel the potential pathways linking environmental factors to health outcomes.

### Co-localization of exposures and impact on CMDs

This study's analysis of the effects of individual environmental exposures is consistent with previous epidemiological findings. For example, studies in China, Europe, and the United States have shown that

smoking increases the risk of adverse outcomes by 1.5- to 3-fold[10,25,26], which is similar in magnitude to the effect estimated in this study. In multiple exposure scenarios, the health effects of a individual environmental factor are influenced by other factors. The Danish study on stroke and myocardial infarction and the Greece study on cardiometabolic mortality showed effects of green spaces that varied with adjustment for $PM_{2.5}$ or noise[12,14,15].

Considering the intercorrelation of exposures, our study discerns co-exposure effect with significant variations in disease risk, by clustering five human-centered exposure patterns. Echoing previous analyses of exposure clustering, in diabetes studies, disease risks in subgroups exposed to high levels of air pollutants like $PM_{2.5}$, $PM_{10}$, and $NO_2$ are notably elevated[27], while those engaged in diet and physical activity fare better[28]. However, these studies typically concentrate on a few exposures within a single environment. Our work extends this to a more realistic multi-dimensional exposure framework, encompassing individual, social, and natural environments. The Social Deprivation subgroup emerges as the highest-risk category across all diseases, characterized by a high prevalence of smoking, elevated IMD scores and social isolation index, and high prevalence of anxiety and depression[8,29-31]. It reinforces the significant contribution of these social stressors, which is supported by our findings on individual exposure contributions.

### Protein signatures and immune-inflammatory responses

The protein signatures varied across the exposure patterns, with marked differences in protein abundance, concentration and

corresponding weights. Notably, enrichment analysis further highlighted the cytokine-cytokine receptor interaction pathway across all four exposure patterns, suggesting a strong link between environmental exposures and immune-inflammatory responses[32,33]. Similarly, multiple pathways, including TNF response, cytokine activity, and monocyte chemotaxis, were enriched, emphasizing the regulatory role of immune responses under multiple environmental factors[32,34,35]. Additionally, extracellular matrix (ECM)-related pathways were enriched in several patterns, indicating that these gene sets are involved in cell migration, tissue repair, and cell-matrix interactions[36,37]. Enrichment of molecular functions like cytokine activity and cytokine binding underscored the significance of immune cell interactions. Our findings align with previous studies, as Perry et al. identified associations of air pollution, climate, and the built environment with proteomic pathways related to angiogenesis, endothelial survival, cell growth, inflammation, and adaptive immunity[24]. Notably, Viral protein interaction with cytokine and cytokine receptor pathways were enriched in the Social Deprivation and Air and Noise pollution patterns[38], suggesting potential involvement of viral infections or their impact on immune responses[39,40]. The osteoclast differentiation and bone metabolism pathways, observed in the Health Behaviors pattern, imply that exercise may protect from bone-related disease[41–43]. Furthermore, the correlation between the computed signature score and the incidence of CMDs is analogous to the correlation between patterns and CMDs incidence, suggesting its potential in capturing physiological responses to exposure. Notably, when adjusted for the corresponding co-exposure pattern, the associations between the proteomic signature scores and health outcomes almost remained unchanged, highlighting its stability and representativeness in reflecting exposure-related health effects. A stable and representative proteomic signature score not only quantifies biological changes induced by exposure but also facilitates further investigation into the etiology of the disease.

## Key proteins mediated CMD risks

In our mediation analysis, we sought to quantify the mediating roles of proteins in the relationships between different patterns and diseases. We identified that CDCP1, CXCL17, FGF21, GDF15, and IGFBP4 play significant roles in mediating the relationship between Social Deprivation and Air and Noise pollution patterns and all CMDs. This suggests that Social Deprivation and Air and Noise pollution patterns may influence the occurrence of CMDs through multiple common biological pathways, which is also consistent with previous studies[24]. Many of the involved molecules are associated with inflammation, metabolic stress, and vascular health, indicating that these pathways may promote the development of CMDs by affecting metabolism, immune response, and vascular function under environmental and social exposures[44]. For example, GDF15 is known to inhibit islet inflammation, oxidative stress and $\beta$ cell apoptosis, thereby reducing the occurrence and development of diabetes. This is typically observed in cases where stressors such as air pollution and socioeconomic status are present[45,46]. Additionally, CXCL17 and FGF21 have been demonstrated to play a role in immune modulation and metabolic dysfunction[47,48]. Animal experiments have shown that air pollution can affect the expression of CXCL17[49], while epidemiologic studies have also found an association between $PM_{2.5}$ and FGF21[50]. Additionally, substantial evidence suggests that exposures like smoking and depression are associated with IGFBP4 and CDCP1[51,52]. These molecules may contribute to CMD progression through their effects on cell migration, metabolic disturbances, and immune responses, highlighting their potential as biomarkers and intervention targets. Modulating the expression or function of these molecules could help alleviate the risks of cardiovascular metabolic diseases induced by environmental pollution and social deprivation. When we examined the proteins mediating the health-promoting environmental exposures, we found a mechanistic difference between the two patterns. In the Blue and Green Spaces pattern, the proteins mediating protective effects against death were ITGAV, HPGDS, ADGRG2, and MMP12. However, the interpretation of protein in the Blue and Green Spaces pattern requires caution. On the one hand, in renal diseases, the protein signature scores inversely correlated with the pattern, suggesting additional explanatory mechanisms beyond these proteins. On the other hand, after adjusting for renal function, only MMP12 retained significant mediation effects. This protein may exert protective effects through immunomodulation and anti-inflammatory pathways, as evidenced by its role in macrophage regulation and extracellular matrix remodeling[53]. CXCL13, CA6, SDC1, and PTN, on the other hand, mediate the relationship between Health Behaviors and renal diseases, diabetes, and death. Previous studies have shown that a high-sugar diet linked to CA6, while a high-fat diet can lead to elevated PTN, and physical activity is associated with CXCL13 and SDC1[54–57]. By modulating the immune response, cellular repair, acid-base balance, and cell-matrix interactions, down-regulation of the proteins may either protect or promote the progression of diabetes and even death[54,58].

## Strength and limitation

This study explored the relationship between various environmental factors and CMDs and identified five co-exposure patterns. Among these, the Social Deprivation pattern, characterized by smoking, low socioeconomic status, social isolation, and mental health issues, was associated with the highest risk of CMDs. This exposure pattern accounted for over 20% of the study population, significantly increasing the CMD burden. Policymakers may consider shifting their focus from individual risk factors to addressing this specific co-exposure pattern and exploring potential measures to support the affected population in alleviating mental health issues and adopting healthier behaviors and lifestyles. Additionally, we investigated the role of proteins in environmental exposure and disease development at the molecular level, deepening our understanding of inflammation and immune-related protein responses triggered by environmental factors. This provided further evidence for future experimental and mechanistic research.

One of the strengths of this study lies in its foundation on the extensive UK Biobank cohort, along with comprehensive Olink proteomics information. The large sample size and comprehensive dataset facilitated thorough confounder control and supported analyses with strong statistical power. We included a wide range of CMD risk factors, covering various levels of environmental exposure. Additionally, CMD data from different organ systems were separately analyzed to account for possible heterogeneity in disease progression. However, several limitations must be considered. First, the cross-sectional data limits causal inference. Although the exposure pattern constructed in this study aimed to capture relatively stable, normative environment over time, the assumption of invariance over time against such a long period of time needs to be approached with caution as the questionnaire information was only collected at baseline. Additionally, there may be reverse causation between proteins and diseases. Due to the insidious nature of CMDs, participants might not exhibit evident clinical symptoms but could already be in the prodromal stage of the disease. We tested the robustness of our findings by conducting a sensitivity analysis, excluding cases with onset within 60 days and within 1 year of the baseline exposure assessment. Second, while LASSO regression is an effective tool for feature selection, it may omit variables with smaller effects or high intercorrelations. This could result in the exclusion of potentially relevant proteins, especially in the context of complex protein profiles. Future research could benefit from incorporating a broader set of proteins and more advanced machine-learning techniques. Additionally, we applied the Bonferroni correction for multiple testing in the primary protein analysis. Although this stringent approach minimizes false positives, it may increase false negatives due to reduced statistical power. Lastly,

functional annotation and enrichment analyses are reliant on existing databases. Consequently, certain proteins may lack comprehensive study or annotation, which constrains the biological interpretation of our findings. Finally, future research could leverage available data—such as epigenomic markers as environmental exposure proxies, protein cis-QTLs, and cardiometabolic GWAS—in Mendelian randomization analyses to strengthen causal validation of environment-protein-disease pathways.

In conclusion, this study identifies exposure patterns and proteomic pathways associated with CMDs within the UK population. The findings provide a scientific foundation for identifying target populations with disadvantaged cardiometabolic health and offer precise guidance for policy interventions. Additionally, this work contributes to understanding the mechanistic links between environmental factors and disease through protein levels.

## Methods

### Study population and ethical compliance
The UK Biobank comprises approximately 500,000 participants aged 37–73 years, who were recruited between 2006 and 2010 from England, Wales, and Scotland. The study protocol was approved by the North West Multi-Center Research Ethics Committee as a Research Tissue Bank, and all participants provided written informed consent, allowing researchers to access and use the data for research purposes. This study was conducted under UK Biobank application ID 117185.

Participants underwent extensive baseline assessments, including questionnaires on sociodemographic factors, lifestyle, and environmental exposures, along with physical measurements and collection of biological samples for long-term storage. A detailed description of the sampling design and data quality of UKB was published elsewhere[59] (Supplementary Methods). In the analysis, participants with missing data or condition before registration were excluded. Details were provided in Supplementary Fig. 6. Participant characteristics, including age, self-reported sex and health condition, are summarized in Supplementary Table 1, Supplementary Table 11 and Supplementary Table 12.

### Exposure assessment
Air pollution measurements include $NO_x$, $NO_2$, $PM_{10}$, and $PM_{2.5}$(Field ID: 24004, 24003, 24005, 24006). These pollutants were estimated for 2010 using a Land Use Regression (LUR) model derived from the European Study of Cohorts for Air Pollution Effects (ESCAPE) project, capturing local environmental exposures across various locations[60] (Supplementary Methods).

Noise pollution (Field ID: 24024) was evaluated based on the average 24-hour sound level for each address in 2010 (Field ID: 24024), modeled through a LUR approach as part of the ESCAPE project.

Residential green and blue spaces estimate reflected the percentage of green areas and water coverage within a 300 m buffer zone around the home (Field IDs:24503, 24505).

Sedentary behavior time was calculated from self-reported daily hours spent watching TV and using a computer (Field IDs: 1070, 1080), with a maximum cap of 24 h per day.

Leisure physical activity was calculated in MET-minutes per week based on baseline frequencies and durations of activities such as walking for pleasure, light and heavy DIY, strenuous sports, and other exercises (Field ID: 971, 981, 1011, 1021, 2624, 2634, 991, 1001, 3637, 3647)[61].

Total physical activity was calculated as MET-minutes per week based on baseline frequency and duration of walking, moderate and vigorous physical activity (Field ID: 864, 874, 884, 894, 904, 914), following IPAQ guidelines[62].

Sleep duration(Field ID:1160) was categorized into two categories based on responses to the questionnaire, with 7 h being considered the optimal sleep[63].

Smoke status was categorized based on responses to the questionnaire (Field ID: 20116) as Never, Current, or Previous smoker.

Diet index based on American Heart Association guidelines[64], includes five factors: vegetables, fruits/day, fish serving, unprocessed red meat and processed meat servings. Scores range from 0 to 5, with higher scores indicating a more health-promoting diet.

IMD score assesses poverty levels in UK local authorities, with variations for England (Field ID: 26410), Scotland (Field ID: 26427), and Wales (Field ID: 26426). It comprises multiple dimensions such as income, employment, health, education, housing access, living environment, and crime.

The social isolation index is calculated by the responses to three questions on the questionnaire (Field ID: 709, 1031, 6160), assigning one point each for living alone, infrequent social visits (less than once a week), and lack of participation in social or leisure activities, resulting in a score range of 0 to 3.

Depression or (and) anxiety was derived from baseline questionnaires. Individuals were classified as depressed if they reported depression (Field ID: 20001) or had a PHQ-4 score above 2 (Field IDs: 2050, 2060, 2070, 2080), and as anxious if they reported anxiety (Field ID: 20001).

### Outcome
Cardiometabolic diseases and related outcomes, which include heart disease, cerebrovascular disease, renal disease, type 2 diabetes, and all-cause death, were classified according to the 10th Edition of the International Classification of Diseases (ICD-10)(Supplementary Methods). Disease records were obtained from Read codes in Primary Care data, Hospital Inpatient data, and Death Register records, using the earliest recorded date across these sources. Data collection ended on different dates depending on the region: October 31, 2022, for England; May 31, 2022, for Wales; and August 31, 2022, for other regions.

### Proteomics profiling
UK Biobank stored baseline plasma samples at −80 °C and in liquid nitrogen (LN2). Blood samples from a randomly selected group of participants were sent to the Olink Analysis Service in Sweden for proteomic profiling using Olink's proximity extension assay, conducted in March 2023. Detailed assay performance and validation information is available in prior publications[65]. The UK Biobank conducted thorough quality control and external validation. After excluding proteins with more than 20% missing values, a total of 2911 proteins were retained for further analysis(Supplementary Data 1). For proteins with missing data under 20%, missing values were imputed using the mean[65,66].

### Covariates
Based on relevant studies, covariates include age, sex, ethnicity, alcohol consumption status, and body mass index (BMI) at baseline. Age and BMI are continuous variables, while other variables are categorical variables. Ethnicity classifications encompassed White, Mixed, Asian or Asian British, Black or Black British, Chinese, and Other. Alcohol consumption status was categorized as Never, Former, Current, or Unknown.

### Statistical analyses
**Effects and contributions of individual exposures.** With the Cox proportional hazards regression model, we estimated the associations between 16 distinct individual exposures and a spectrum of five CMDs. For each CMD, we developed 16 single-exposure models, each specific to one exposure, alongside a comprehensive multi-exposure model that combined all exposures, thereby exploring potential interactive effects. All models were all adjusted for sex, age, ethnicity, alcohol consumption status and BMI to account for confounding variables.

The *p* value in single-exposure models was adjusted with Bonferroni method. Subsequently, we trained eXtreme Gradient Boosting (XGBoost) model to handle potential complex nonlinear and interactive effects between multiple exposures and diseases. For each disease model, we trained a regression framework that incorporated all exposures and confounding variables (sex, age, ethnicity, alcohol consumption status and BMI) as inputs to forecast the hazard of disease. We set the maximum tree depth to 3 and the learning rate to 0.03, allocated 70% of the data to training, and executed 1,000 boosting iterations to optimize predictions. To evaluate model performance, we computed the C-index and generated receiver operating characteristic (ROC) curves in the test datasets. Using the XGBoost models, we further calculated SHapley Additive exPlanations (SHAP) values for each exposure, quantifying the individual exposure contributions to the model's predictions.

**Cluster exposure pattern.** The K-prototype method was employed to identify exposure patterns based on 16 exposures, which include continuous, multiple categorical, and binary categorical variables. The K-prototype algorithm, a combination of K-means and K-modes, is well-suited for clustering mixed attribute data. The clustering criterion involves using an appropriate loss function to measure the distance between both numerical and categorical variables and the prototype. To determine the optimal number of clusters, we applied the Elbow method with the CH index. The change in the CH index flattened out at five clusters, indicating that this number provided a balanced clustering solution (Supplementary Fig. 7). In this study, cluster analysis was only performed in samples with complete exposure data to ensure the accuracy of the results. We assessed the association between the identified patterns and CMDs using Cox proportional hazards models, adjusting for sex, age, ethnicity, alcohol consumption status, and BMI.

**Proteomics analysis.** Within each exposure pattern subgroup, we used the Cox proportional hazards models to assess the effect of 2911 proteins on CMDs, adjusting for sex, age, ethnicity, alcohol consumption status, and BMI. Based on the existence of systematic associations in the pathogenesis of CMDs, we selected proteins that were statistically significant (adjusted *p* value < 0.05) across all diseases within each subgroup, which were subsequently pooled for further signature analysis. To avoid an excessively conservative selection of such intersections, we subsequently added a sensitivity analysis.

LASSO regression (glmnet R package) was facilitated to filter key proteomic biomarkers for each exposure pattern. The penalty parameter (λ) was optimized using 10-fold cross-validation. Given the categorical nature of the exposure patterns, a regularized multinomial regression model was applied, incorporating maximum multinomial likelihood and item-specific penalties. KEGG and GO enrichment analyses (clusterProfiler R package) were conducted to investigate the biological pathways and processes associated with the genes targeted by these feature proteins. To facilitate interpretation, the reference pattern was assigned a coefficient of 0, and the coefficients for other patterns were adjusted to represent differences relative to the reference. The proteomic signature score for each exposure pattern was then computed as a weighted sum of the selected feature proteins. Furthermore, Cox proportional hazards models assessed the association of proteomic signature scores with disease risk, reporting HR and 95%CI.

In addition, we applied the mediation analysis of the each protein selected by LASSO regression, adjusting for sex, age, ethnicity, alcohol consumption status, and BMI (CMAverse R package). Specifically, the mediation model is a linear model based on the relationship between one co-exposure pattern and one protein, and the outcome model is a Cox proportional hazards model of an exposure pattern, a protein, and a disease. We estimated the natural total indirect effect and mediated proportion through the regression-based approach, and the *P* values

were adjusted by Bonferroni method within each exposure pattern-outcome pair. Considering that inconsistent mediation would lead to unstable PM estimates, all conclusions were drawn when indirect and direct effects were in same direction.

The sensitivity analysis included the following: First, to examine the assumption of linearity for continuous confounders, we conducted Cox proportional hazards models using categorized age and BMI. Specifically, age was categorized as < 60 and ≥60 years, and BMI as underweight ($< 18.5 \, \text{kg/m}^2$), normal ($18.5–24.9 \, \text{kg/m}^2$), overweight ($25.0–29.9 \, \text{kg/m}^2$), and obesity ($≥30 \, \text{kg/m}^2$). Second, for individual exposure contribution, we changed the machine learning parameter Settings and compared the C-index. Third, we assessed the associations between co-exposure patterns and CMDs, as well as between protein signatures and CMDs, within participants without pre-existing medical conditions at baseline to evaluate robustness. Fourth, we explored the association between protein signature scores and CMDs in datasets that excluded the ones who experienced diseases within 60 days and within one year, respectively, for robust profiles of protein signatures. Fifth, we further adjusted for serum creatinine (as the measure of renal function) in Cox proportional hazards models to assess the associations of proteomic signature scores with CMDs. Finally, we conducted mediation analyses for all proteins, additionally adjusting for creatinine.

**Statistics and reproducibility.** This study is an observational study based on participants recruited from the UK Biobank cohort. No statistical method was used to predetermine sample size. are described in the Study Population and Ethical Compliance section and detailed in Supplementary Fig. 6. No data were excluded from the analyses beyond the predefined inclusion/exclusion criteria. Traditional experimental design elements were not applicable: the analyses were not randomized, and the investigators were not blinded to allocation or outcome assessment.

### Reporting summary
Further information on research design is available in the Nature Portfolio Reporting Summary linked to this article.

## Data availability
This study is based on data from the UK Biobank (Application Number 117185). Detailed information about data availability in UK Biobank can be accessed online at www.ukbiobank.co.uk. Source data are provided with this paper.

## Code availability
Codes for the analyses are available at (https://github.com/Vi-1105/ExposurePatternCMD/tree/main).

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

## Acknowledgements

This research analysed data provided by the UK Biobank. We thank all participants and their families, as well as all the investigators and members of the UK Biobank. We appreciate the resources and the platform provided by West China School of Public Health and West China Fourth Hospital of Sichuan University. H.X. was supported by the Noncommunicable Chronic Diseases-National Science and Technology Major Project (2024ZD0542200), the Postdoctoral Fellowship Program of the China Postdoctoral Science Foundation (CPSF) (GZB20250193). X.X. was supported by the National Natural Science Foundation of China (82273740). J.Z. was supported by the National Natural Science Foundation of China (82073667). B.G. was supported by the National Natural Science Foundation of China (82103943). The funders had no role in the study design or implementation; data collection, management, analysis, or interpretation; manuscript preparation, review, or approval; or the decision to submit the manuscript for publication.

## Author contributions

X.T., H.X. and X.Z. were responsible for the study concept and design. X.T., H.X. and G.W. performed statistical analyses and drafted the manuscript. S.Y. and Y.Y. checked the statistical analysis. Y.A., Y.J., Z.W. and J.Z. proofread the manuscript. S.C., X.X., B.G. and Z.X. critically revised the manuscript. H.X., J.Z. and B.G. obtained funding. All authors contributed to the acquisition or interpretation of data, proofreading of the manuscript for important intellectual content and the final approval of the version to be published. The corresponding author attests that all listed authors meet authorship criteria and that no others meeting the criteria have been omitted.

## Competing interests

The authors declare no competing interests.
