## [Transparent Peer Review file · Nature Communications]

Personal, Social, and Natural Co-Exposure Pattern and Plasma Proteins in Cardiometabolic Diseases

Corresponding Author: Dr Xing Zhao

Version 0:

Reviewer comments:

Reviewer #1

(Remarks to the Author)

This work explores a highly relevant and timely topic in public health, investigating how environmental exposures cluster in communities and their contribution to CMDs. The authors employ large-scale proteomic data to examine whether environmental exposures impact human health through protein pathways. Given the limited number of published studies on this topic, particularly those integrating proteomics data, this paper holds scientific importance and has the potential to provide novel insights to the research community. However, there are some areas where the causal inference chain could be strengthened, which I encourage the authors to address to enhance the robustness of their findings.

Narratives:

- o The abstract would benefit from including key details such as the study population, sample size, statistical methods, and key results (statistics). This would provide readers with a clearer overview of the study's scope and quantitative findings.
- o The authors state in the Introduction section that 'Although the role of proteomics in disease diagnosis is widely recognized, it remains unclear whether protein signatures reflect environmental exposures in physiological responses. The extent to which feature proteins explain the effect of environmental exposures on CMDs, which could reveal key biological pathways and mechanisms, has also been poorly understood' While I appreciate this as a key motivation for the study, it is worth noting that some prior research has explored the relationship between proteomics and environmental exposures, such as the study by Perry et al. (Proteomics, Human Environmental Exposure, and Cardiometabolic Risk). The authors may wish to refine this statement to better reflect the existing literature.
- o When discussing co-exposures in real-world settings, the authors cite only one animal model study. However, there are relevant studies involving human participants, such as Kasdagli et al. (Environ Pollut. 2022;292(Pt B):118372), which could be included to strengthen the background.
- o On line 69, the term "incident rate" is used inappropriately, as it should not be expressed as a percentage. I recommend revising this and carefully reviewing other instances where "rate" is used similarly.
- o The authors state that 'For instance, individuals in the "Air and Noise Pollution" group had a 1.056 times higher hazard of heart disease than those in the reference group (95% CI: 1.012–1.101). I assume this is intended to convey that the hazard ratio (HR) for individuals in the 'Air and Noise Pollution' group is 1.056, indicating a 5.6% higher risk of developing heart disease compared to the reference group. This clarification would improve readability.
- o Hazard ratios are typically reported with two decimal places, though I acknowledge this may depend on journal-specific formatting requirements.
- o The font size and resolution of Figure 4 are too small to be clearly visible. Additionally, table captions are missing in the supplementary tables. I recommend revising these elements to improve accessibility and clarity.
- o The authors state that 'The associations between proteomics signatures for each co-exposure pattern and CMDs were strongly aligned with the CMDs correlations observed for each co-exposure pattern'. However, there appears to be a discrepancy between the 'green and blue spaces' exposure pattern and its reflection in proteomic signatures in relation to renal disease risks. The authors may wish to elaborate on this observation.
- o In Supplementary Figure 4, the authors adjust for co-exposure patterns when investigating proteomic signatures and disease outcomes. It would be helpful to provide a clearer rationale for this adjustment.
- o The authors discuss the interventional utility of proteins for CMD outcomes in several sections. While I agree that proteins with mediating effects are valuable for understanding mechanisms underlying environmental co-exposures in CMD pathophysiology, I caution against overemphasizing their interventional utility. Interventions targeting environmental

exposures (e.g., reducing air pollution or increasing green spaces) may be more intuitive and effective than targeting mediators.

o The Discussion could be strengthened by including insights on how CMD risks can be mitigated in practice, both at individual and societal levels. This would enhance the translational relevance of the findings.

Methodology:

o Studies have shown that circulating protein levels are significantly influenced by kidney function. Did the authors adjust for kidney function data, or do they have justifications for not doing so? Clarifying this would strengthen the methodology.

o How many individuals with prevalent diseases were included in the study, and how were these samples treated in the analyses? Providing this information would improve transparency.

o The authors stated that 'Proteins that were statistically significant (adjusted P-values < 0.05) across all diseases within each subgroup were selected and subsequently pooled for further signature analysis'. Given that proteomic profiles may differ between diseases and exposure patterns, could the authors elaborate on why only proteins associated with all outcomes in each group were selected? This approach may warrant further justification.

o The authors stated that 'P-value was adjusted by Bonferroni method.' Please clarify how this adjustment was applied—for example, was it performed per exposure pattern-outcome pair or across all comparisons?

o The mediation analysis and the construction of the relationship between exposure patterns and proteins require careful consideration. The authors aim for causal inference, as stated in the Methods: 'The regression-based causal mediation analysis was carried out to fit the mediation model and the outcome model respectively.' While the use of time-to-event analysis to establish causal relationships between exposure patterns and outcomes, and between protein scores and outcomes, is appropriate, the use of LASSO regression for exposure pattern-protein relationships is cross-sectional and cannot infer causality. Additionally, the description of how indirect and direct effects were calculated is unclear. If the indirect effect was calculated by multiplying the LASSO estimate with the Cox regression estimate, this approach may present methodological challenges. I encourage the authors to address these points in detail.

o In the mediation analysis, the proportion mediated by several proteins exceeds 100%, with the direct effect being larger than the total effect. This finding warrants careful investigation and interpretation, as it may indicate potential issues with the mediation model.

o Finally, while it may be beyond the scope of this manuscript, it is worth noting that leveraging publicly available data—such as epigenome data for environmental exposures, cis-expression quantitative trait loci (cisQTLs) for proteins, and genome-wide association studies (GWAS) for CMDs—to conduct Mendelian randomization (MR) analyses could provide more robust evidence for causal inference.

Reviewer #2

(Remarks to the Author)

General comment:

Interesting MS. I have a number of comments, however, that may help to improve the work. Especially the biological and pathophysiological relevance of the identified biomarkers should be better explained and presented. Also the major underlying concept, the exposome, is even not mentioned here and many landmark publications are not cited and discussed – a clear weakness of the manuscript.

Specific comments

1) The exposome concept provides the fundamental basis for the present studies. The authors should explain the exposome concept in detail and also cite respective landmark papers on this concept (e.g. PMID: 16103423, PMID: 23519765, PMID: 24659601, PMID: 37165157).

2) Landmark studies on multi-exposure scenarios (PM, UFP, NO_x, noise, green space) with respect to cardiometabolic risk should be discussed in detail and respective references must be cited (PMID: 36334460), also published for stroke (PMID: 37265507) and MI (PMID: 37738461) - also graphically summarized in PMID: 37165157.

3) Previously, the tight interconnection of different environmental and lifestyle exposures was reviewed in detail, also based on examples for co-localization of major exposure risk factors in the USA (PMID: 34005032).

4) It would be important if the authors can associate their top candidates of modulated proteins by the different environmental risk factors (e.g. CDCP1, CXCL17, FGF21, GDF15 and IGFBP4 for "Air and Noise Pollution" and "Social Deprivation"; for "blue and green spaces" and "health behaviors" ITGAV, HPGDS, ADGRG2 and MMP12 for the former and CXCL13, CA6, SDC1 and PTN for the latter) with some previous published reports – either animal or human – where these markers were also found modulated by specific exposures.

5) Figure 2: When looking at the hazard ratios for the different environmental & lifestyle risk factors a certain discrepancy between the present findings and the reports by the GBD study become evident. In the GBD air pollution is meanwhile a leading driver of global deaths and burden of disease – just at the same level as tobacco smoking. This is not reflected at all by the present data. How can this be explained?

6) Since these are data from the UK Biobank I assume that recruitment was done in the UK? Please provide a map providing information on the regional distribution of the recruited study population (can be provided in the supplement).

7) Exposure maps must be provided for the following environmental risk factors (can be presented in the supplement): NO_x, NO₂, PM₁₀, PM_{2.5}, Noise, green space, blue space.

8) A summarizing scheme would be helpful to highlight the major (patho)physiological pathways triggered by the different top candidates for the specific exposures. This would allow the reader to better understand the individual pathomechanisms triggered by the different exposures.

Version 1:

Reviewer comments:

Reviewer #1

(Remarks to the Author)

The authors have conducted a thorough revision to response my comments, which is greatly appreciated. The paper now has been substantially improved.

I agree with the authors' response regarding the inappropriateness of using GWAS of environmental exposures for Mendelian randomization analysis, and that was why I mentioned the 'epigenome data for environmental exposures' instead of 'GWAS data of environmental exposures'. Even though the MR results were under expectation, I suggest to remove the MR part of the study but simply mention it in the discussion as future investigations.

Reviewer #2

(Remarks to the Author)

No further comments

Reviewer #3

(Remarks to the Author)

Thanks to the author team for their work. This is indeed an important piece of work, and I think the authors have addressed all reviewers' comments. I'm happy with those modifications.

However, a few minor comments from me:

1. In lines 446–448, please refer to existing literature to support the method you use to impute the missing data.
2. In the Cox proportional hazards regression model from your "Effects and contributions of individual exposures" section, you adjusted for BMI and age as continuous variables. That is based on the hypothesis that the effects of age and BMI are linear. Please consider using a non-linear method to adjust these confounders—e.g., polynomial terms or categorical variables.

Version 2:

Reviewer comments:

Reviewer #3

(Remarks to the Author)

I'm happy with those modifications. No further comments

made.

Dear Reviewers,

Thanks very much for providing us a chance to revise our manuscript. We appreciate all the valuable comments on our work from the reviewers and have responded to all the comments from the reviewers as requested. We hope that this revised submission meets the reviewers' expectations.

REVIEWER COMMENTS

Reviewer #1 :

This work explores a highly relevant and timely topic in public health, investigating how environmental exposures cluster in communities and their contribution to CMDs. The authors employ large-scale proteomic data to examine whether environmental exposures impact human health through protein pathways. Given the limited number of published studies on this topic, particularly those integrating proteomics data, this paper holds scientific importance and has the potential to provide novel insights to the research community. However, there are some areas where the causal inference chain could be strengthened, which I encourage the authors to address to enhance the robustness of their findings.

Response:

Thank you very much for your thoughtful review of our manuscript. Your valuable comments and suggestions have greatly enhanced the quality and scientific rigor of our work. We have diligently incorporated your constructive suggestions into our manuscript revisions. Please find details in the point-by-point response below. **Please note that the line numbers mentioned in the following responses correspond to those in the revised manuscript.**

Narratives:

1. The abstract would benefit from including key details such as the study population, sample size, statistical methods, and key results (statistics). This would provide readers with a clearer overview of the study's scope and quantitative findings.

Response:

Thanks for your suggestion. We have revised the abstract to include key details such as the study population, sample size, statistical methods, and main quantitative findings.

2. The authors state in the Introduction Section that 'Although the role of proteomics in disease diagnosis is widely recognized, it remains unclear whether protein signatures reflect environmental exposures in physiological responses. The extent to which feature proteins explain the effect of environmental exposures on CMDs, which could reveal key biological pathways and mechanisms, has also been poorly understood' While I appreciate this as a key motivation for the study, it is worth noting that some prior research has explored the relationship between proteomics and environmental exposures, such as the study by Perry et al. (Proteomics, Human Environmental Exposure, and Cardiometabolic Risk). The authors may wish to refine this statement to better reflect the existing literature.

Response:

Regarding the study by Perry et al., we recognize it as an important theoretical foundation that investigates the relationships among environmental factors (built environment, green space, air pollution, temperature, and indicators of social vulnerability), proteins, and cardiometabolic and respiratory outcomes. In the revised manuscript, we have added relevant citations and revised the Introduction Section to more clearly articulate that our study aims to explore the protein signatures of environmental co-exposure patterns—a departure from previous research directions. Additionally, we have incorporated this research information into the Discussion Section. Specifically, the revisions include:

Introduction:

Plasma proteins, in addition to serving as therapeutic and predictive disease biomolecules^{20, 21}, can provide insights into the intrinsic mechanisms associated with environmental exposures to human health conditions and cardiometabolic diseases^{22, 23}. According to the findings of previous studies, which explored its function at the molecular level in the pathogenesis of various environmental exposures²⁴, the question remains unanswered whether it can similarly reflect the physiological responses to co-exposure patterns and whether it can reveal key biological pathways and mechanisms of the effect on CMDs. (Lines 47-55)

Discussion:

Notably, enrichment analysis further highlighted the cytokine-cytokine receptor interaction pathway across all four exposure patterns, suggesting a strong link between environmental exposures and immune-inflammatory responses^{32,33}. Similarly, multiple pathways, including TNF response, cytokine activity, and monocyte chemotaxis, were enriched, emphasizing the regulatory role of immune responses under multiple environment^{34,35,36}. Additionally, extracellular matrix (ECM)-related pathways were enriched in several patterns, indicating that these gene sets are involved in cell migration, tissue repair, and cell-matrix interactions^{37, 38}. Enrichment of molecular functions like cytokine activity and cytokine binding underscored the significance of immune cell interactions. Our findings align with previous studies, as Perry et al. identified associations of air pollution, climate, and the built environment with proteomic pathways related to angiogenesis, endothelial survival, cell growth, inflammation, and adaptive immunity²⁴. (Lines 239-252)

This suggests that "Social Deprivation" and "Air and Noise pollution" patterns may influence the occurrence of CMDs through multiple common biological pathways, which is also consistent with previous studies²⁴.(Lines 273-276)

3. When discussing co-exposures in real-world settings, the authors cite only one animal model study. However, there are relevant studies involving human participants, such as Kasdagli et al. (Environ Pollut. 2022;292(Pt B):118372), which could be included to strengthen the background.

Response:

We sincerely thank you for pointing out this important literature omission. We fully acknowledge the significant contribution of Kasdagli et al., whose work on the potential synergistic effects of air pollution and greenness on disease risk in the Greek population provides valuable theoretical support for our study. In response, we have incorporated this citation along with additional relevant epidemiological literature in the Introduction Section and have made corresponding revisions in the Discussion Section.

Discussion:

In multiple exposure scenarios, the health effects of a individual environmental factor are influenced by other factors. The Danish study on stroke and myocardial infarction and the Greece study on cardiometabolic mortality showed effects of green spaces that varied with adjustment for PM_{2.5} or noise^{12,15,14}. (Lines 216-220)

4. On line 69, the term “incident rate” is used inappropriately, as it should not be expressed as a percentage. I recommend revising this and carefully reviewing other instances where “rate” is used similarly.

Response:

Thank you for your attention to detail, and we apologize for the inaccuracy. We have revised this term to correctly express as “cumulative incidence”. Additionally, we have thoroughly reviewed the manuscript to ensure that all instances of "rate" are used accurately.

5. The authors state that ‘For instance, individuals in the "Air and Noise Pollution" group had a 1.056 times higher hazard of heart disease than those in the reference group (95% CI: 1.012–1.101). I assume this is intended to convey that the hazard ratio (HR) for individuals in the ‘Air and Noise Pollution’ group is 1.056, indicating a 5.6% higher risk of developing heart disease compared to the reference group. This clarification would improve readability.

Response:

Thank you for your professional correction on the presentation of the results. We fully agree with your suggestion and have refined the risk ratio description in the Result Section as follows:

For instance, individuals in the "Air and Noise Pollution" group had an HR of 1.06 for heart disease, indicating a 6% higher risk compared to the reference group. Elevated HRs were also observed for cerebrovascular disease, renal disease, diabetes, and death, ranging from 1.06 to 1.19. (Lines 116- 120)

6. Hazard ratios are typically reported with two decimal places, though I acknowledge this may depend on journal-specific formatting requirements.

Response:

Thanks for your suggestion, and we have revised the manuscript to ensure that all HRs and other relevant numerical results are consistently reported with two decimal places.

7. The font size and resolution of Figure 4 are too small to be clearly visible. Additionally, table captions are missing in the supplementary tables. I recommend revising these elements to improve accessibility and clarity.

Response:

Thank you for your valuable feedback on the presentation of the figures. We have revised **Figure 4** to ensure higher resolution and appropriately sized fonts. Since the original figure contained excessive information, we have moved the details on protein concentrations for each co-exposure pattern to **Supplementary Table 5**. Additionally, we have carefully added the titles of the supplementary tables on the top to enhance clarity and readability.

Fig.4 Proteomic signature of co-exposure pattern. Left: Feature proteins for each exposure pattern and their corresponding weights were obtained using LASSO regression. Right: Gene Ontology (GO) enrichment analysis was performed on the feature proteins to identify the associated biological processes.

8. The authors state that ‘The associations between proteomics signatures for each co-exposure pattern and CMDs were strongly aligned with the CMDs correlations observed for each co-exposure pattern’. However, there appears to be a discrepancy between the ‘green and blue spaces’ exposure pattern and its reflection in proteomic signatures in relation to renal disease risks. The authors may wish to elaborate on this observation.

Response:

Thanks for your insightful comments. These differential results are not contradictory. The protective effect of "Blue and green space" exposure on renal disease has been reported in previous literature^{1,2}, which is consistent with the results of the present study (Figure 3). As reported in previous studies, this effect may work through various mechanisms, including DNA methylation, metabolism, and gut microbiota³⁻⁵, and also may through protein as explored in this study. However, we did not find significantly negative association between “blue and green space” related proteomic signature and renal disease, which indicated that the protective effect of “Blue and green space” against renal disease may not work through protein. This conjecture is supported by our mediation analysis with no protein identified as significant mediators in the association. To ascertain this, future research is anticipated to explore omics characteristics that can better characterize the impact of "Blue and green space" on renal health.

In the revised manuscript, we have included a detailed description of this discrepancy phenomenon in the **Result and Discussion Sections**.

Result :

The effect of proteomic signatures on CMDs demonstrated alignment with the correlations observed between co-exposure patterns and CMDs risk (Supplementary Fig. 4), particularly through consistent risk elevations seen in the "Air and Noise Pollution" and "Social Deprivation" patterns. The observed alignment was attenuated in renal disease analyses, where the protective effect of "Blue and Green Spaces" differed from that of the proteomic signature.(Lines 153-159)

Discussion :

However, the interpretation of protein in the "Blue and Green Spaces" pattern requires caution. On the one hand, in renal diseases, the protein signature scores inversely correlated with the pattern, suggesting additional explanatory mechanisms beyond these proteins. On the other hand, after adjusting for renal function, only MMP12 retained significant mediation effects. This protein may exert protective effects through immunomodulation and anti-inflammatory pathways, as evidenced by its role in macrophage regulation and extracellular matrix remodeling⁵⁴.(Lines 298-306)

References:

1. Liu M, Ye Z, et al. Relations of residential green and blue spaces with new-onset chronic kidney disease. *Sci Total Environ*. 2023

2. Lee W, Heo S, et al. Associations between greenness and kidney disease in Massachusetts: The US Medicare longitudinal cohort study. *Environ Int.* 2023 Mar;173:107844.
 3. Liu L, Yan LL, et al. Air pollution, residential greenness, and metabolic dysfunction biomarkers: analyses in the Chinese Longitudinal Healthy Longevity Survey. *BMC Public Health.* 2022 May 4;22(1):885.
 4. Jeong A, Eze IC, et al. Residential greenness-related DNA methylation changes. *Environ Int.* 2022 Jan;158:106945.
 5. Wu K, Guo B, et al. Association between residential greenness and gut microbiota in Chinese adults. *Environ Int.* 2022 May;163:107216.
9. In Supplementary Figure 4, the authors adjust for co-exposure patterns when investigating proteomic signatures and disease outcomes. It would be helpful to provide a clearer rationale for this adjustment.

Response:

Thanks for your suggestion. We constructed proteomic signature scores to capture cumulative proteomic changes associated with co-exposure patterns. When adjusted for the corresponding co-exposure pattern, the association between the proteomic signature score and health outcomes changed little, indicating its stability and representativeness of the health effects of exposure. Stable and representative proteomic signature scores not only quantify exposure-induced biological changes, but also provide valuable insights for chronic disease prevention and lifespan extension^{1,2}. We revised the **Discussion Section** to express this point more clearly :

Discussion :

Furthermore, the correlation between the computed signature score and the incidence of CMDs is analogous to the correlation between patterns and CMDs incidence, suggesting its potential in capturing physiological responses to exposure. Notably, when adjusted for the corresponding co-exposure pattern, the associations between the proteomic signature scores and health outcomes almost remained unchanged, highlighting its stability and representativeness in reflecting exposure-related health effects. A stable and representative proteomic signature score not only quantifies biological changes induced by exposure but also facilitates further investigation into the etiology of the disease.(Lines 258-267)

References:

1. Zhu K, Li R, Yao P, et al. Proteomic signatures of healthy dietary patterns are associated with lower risks of major chronic diseases and mortality. *Nat Food.* 2025 Jan;6(1):47-57.
2. Ran S, Zhang J, et al.. Association of metabolic signatures of air pollution with MASLD: Observational and Mendelian randomization study. *J Hepatol.* 2024 Sep 28:S0168-8278(24)02573-X.

10. The authors discuss the interventional utility of proteins for CMD outcomes in several Sections. While I agree that proteins with mediating effects are valuable for understanding mechanisms underlying environmental co-exposures in CMD pathophysiology, I caution against overemphasizing their interventional utility. Interventions targeting environmental exposures (e.g., reducing air pollution or increasing green spaces) may be more intuitive and effective than targeting mediators.

Response:

Thank you for your careful review of our paper and pertinent suggestions. we agree with the reviewers' suggestion that the emphasis on the utility of protein interventions will be weakened and have revised the manuscript accordingly.

Discussion :

These findings not only provide new insights into the multi-dimensional impact of environmental and social exposures on CMDs, but also, through protein analysis, further unravel the complexity of the etiological links between environment and health, enabling quantitative assessment of the health effects of complex environmental exposures using biological process responses.(Lines 205-210)

11. The Discussion could be strengthened by including insights on how CMD risks can be mitigated in practice, both at individual and societal levels. This would enhance the translational relevance of the findings.

Response:

We are very grateful to the reviewers for this constructive comment. In the revised manuscript, we discuss the practical implications of exposure pattern analysis and protein analysis, respectively.

Discussion :

This study explored the relationship between various environmental factors and CMDs and identified five co-exposure patterns. Among these, the "Social Deprivation" pattern, characterized by smoking, low socioeconomic status, social isolation, and mental health issues, was associated with the highest risk of CMDs. This exposure pattern accounted for over 20% of the study population, significantly increasing the CMD burden. Policymakers may consider shifting their focus from individual risk factors to addressing this specific co-exposure pattern and exploring potential measures to support the affected population in alleviating mental health issues and adopting healthier behaviors and lifestyles. Additionally, we investigated the role of proteins in environmental exposure and disease development at the molecular level, deepening our understanding of inflammation and immune-related protein responses triggered by environmental factors. This provided further evidence for future experimental and mechanistic research. (Lines 315- 328)

Methodology:

1. Studies have shown that circulating protein levels are significantly influenced by kidney function. Did the authors adjust for kidney function data, or do they have justifications for not doing so? Clarifying this would strengthen the methodology.

Response:

We appreciate this critical methodological insight. Considering that renal function may be a potential mediator between proteins and CMDs¹, we did not adjust for renal function in the primary analysis. Taking into account the reviewers' comments, we have implemented two sensitivity analyses:

(1) Cox models with additional adjustment for serum creatinine to assess the association between Proteomic Signatures and CMDs: We found that after renal function adjustment, all associations between proteomic signatures and CMDs were essentially unchanged, except for the relationship between "blue and green space" and renal disease, which was no longer statistically significant (Figure R1; Supplementary table 999).

(2) Full mediation analysis with additional adjustment for serum creatinine: Almost key proteins mediated CMDs retained their significant mediating roles, with the exception of ITGAV, HPGDS, and ADGRG2 in the "blue and green space" related association.

We have revised the manuscript by adding the sensitivity analysis described above in the Method Section and the Discussion Section of the results, with a special focus on the effect of renal function.

Method:

Fourth, we further adjusted for serum creatinine (as the measure of renal function) in Cox proportional hazards models to assess the associations of proteomic signature scores with CMDs. Finally, we conducted mediation analyses for all proteins, additionally adjusting for creatinine. (Lines 538-542)

Discussion :

However, the interpretation of protein in the "Blue and Green Spaces" pattern requires caution. On the one hand, in renal diseases, the protein signature scores inversely correlated with the pattern, suggesting additional explanatory mechanisms beyond these proteins. On the other hand, after adjusting for renal function, only MMP12 retained significant mediation effects. This protein may exert protective effects through immunomodulation and anti-inflammatory pathways, as evidenced by its role in macrophage regulation and extracellular matrix remodeling⁵⁴.(Lines 298-306)

References:

1. Cui, C., Liu, L., et al. Triglyceride-glucose index, renal function and cardiovascular disease: a national cohort study. *Cardiovasc Diabetol.* 2023 Nov 28;22(1):325.

Figure R1 Renal Function-adjusted Association Between Proteomic Signatures and Cardiometabolic Diseases

- How many individuals with prevalent diseases were included in the study, and how were these samples treated in the analyses? Providing this information would improve transparency.

Response:

We appreciate the reviewers' comments on the inclusion and treatment of individuals with prevalent diseases. We have included the numbers in **Supplementary Table 15 and Supplementary Table 16**.

To ensure representative exposure patterns in our cluster analysis, we retained individuals with prevalent diseases. We excluded those with the pre-existing disease in correspondent disease-specific analyses (Supplementary Figure 5). Given the potential interrelationships between diseases—where some may act as mediators for others—we did not exclude individuals with other pre-existing conditions, nor did we adjust for other diseases in our models.^{1,2} However, we understood the reviewers' concerns and therefore added a sensitivity analysis that was limited to individuals without any medical history. The results were shown to be consistent with the primary analysis, further supporting the robustness of the study findings. We have revised the **Result and Method Section**.

Supplementary Table 15 Prevalence of Pre-existing diseases in exposure pattern analysis, n(%)

Analysis data set	Heart disease	Cerebrovascular disease	Renal Disease	Diabetes
Heart	0(0)	4206(1.25)	3790(1.13)	13830(4.11)
Cerebrovascular	20691(5.86)	0(0)	4679(1.32)	16593(4.7)
Renal	20903(5.91)	5307(1.5)	0(0)	16419(4.64)
Diabetes	18580(5.44)	4858(1.42)	4056(1.19)	0(0)
Death	22093(6.16)	5608(1.56)	4980(1.39)	17343(4.83)

Supplementary Table 16 Prevalence of Pre-existing diseases in proteomic analysis, n(%)

Analysis data set	Heart disease	Cerebrovascular disease	Renal Disease	Diabetes
-------------------	---------------	-------------------------	---------------	----------

Heart	0(0)	470(1.34)	509(1.45)	1494(4.26)
Cerebrovascular	2439(6.59)	0(0)	670(1.81)	1832(4.95)
Renal	2416(6.54)	608(1.64)	0(0)	1788(4.84)
Diabetes	2192(6.13)	561(1.57)	579(1.62)	0(0)
Death	2631(6.98)	662(1.76)	724(1.92)	1933(5.13)

Method:

Second, we assessed the associations between co-exposure patterns and CMDs, as well as between protein signatures and CMDs, within participants without pre-existing medical conditions at baseline to evaluate robustness. (Lines 532-535)

Result:

Second, the relationship between exposure patterns and diseases within individuals who had no pre-existing CMDs at baseline were largely consistent with relationship in whole population: "Air and Noise Pollution" and "Social Deprivation" had significant adverse effects on CMDs, while "Health Behaviors" and "Blue and Green Space" exhibited protective effects (Supplementary Table 10). The relationships between protein signature and diseases in the subgroups also remained consistent (Supplementary Table 11). (Lines 175-182)

References:

1. Spence JD, Urquhart BL. Cerebrovascular Disease, Cardiovascular Disease, and Chronic Kidney Disease: Interplays and Influences. *Curr Neurol Neurosci Rep.* 2022 Nov;22(11):757-766.
2. Ndumele CE, Neeland IJ, et al. A Synopsis of the Evidence for the Science and Clinical Management of Cardiovascular-Kidney-Metabolic (CKM) Syndrome: A Scientific Statement From the American Heart Association. *Circulation.* 2023 Nov 14;148(20):1636-1664.
3. The authors stated that ‘Proteins that were statistically significant (adjusted P-values < 0.05) across all diseases within each subgroup were selected and subsequently pooled for further signature analysis’. Given that proteomic profiles may differ between diseases and exposure patterns, could the authors elaborate on why only proteins associated with all outcomes in each group were selected? This approach may warrant further justification.

Response:

We sincerely appreciate the reviewer's astute observation. Our selection of proteins showing statistical significance across all subgroup diseases (p-adjusted<0.05) was predicated on the following considerations: the interSection approach aligns with our study's primary hypothesis that cardiometabolic diseases shares similar systemic pathophysiological mechanisms^{1,2,3}, which should manifest as proteins consistently dysregulated across all disease states.

To address the reviewer's concern about potentially overlooking disease-specific mediators, we added the following complementary analyses.

(1) Expanded LASSO regression using the union set of proteins associated with any diseases (n=2,040) : The result concordant with our original findings. The union-derived signature

shows minimal changes in both the magnitude order and directionality of the protein weights compared to the original interSection-derived signature (Figure R2). The pathway enrichment analysis revealed a substantial overlap in the significantly enriched biological processes between the interSection- and union-derived signatures (Figure R3).

(2) Full mediation analysis for all 2,911 proteins: Fundamentally, the distinction between interSection and union approaches lies primarily in candidate pool selection. Analyzing all the protein avoids this selection. As described in our response to Methodology Q1, the result essentially maintained the original conclusions (Supplementary Table 999)

These complementary analyses collectively suggest that while the interSection approach provides more conservative estimates, the core biological insights remain robust to alternative protein selection criteria. We have expanded the **Method Section** to explicitly discuss this methodological consideration and added corresponding Sensitivity Results to **Supplementary Material**.

Method:

Based on the existence of systematic associations in the pathogenesis of CMDs, we selected proteins that were statistically significant (adjusted p-value < 0.05) across all diseases within each subgroup, which were subsequently pooled for further signature analysis. To avoid an excessively conservative selection of such intersections we subsequently added a sensitivity analysis.. (Lines 499-504)

Finally, we conducted mediation analyses for all proteins, additionally adjusting for creatinine. (Line 541-542)

Result:

Fifth, mediation analyses were performed on all proteins. With the exception of ITGAV, HPGDS, and ADGRG2 for "Blue and Green space", the "Air and Noise pollution", "Social Deprivation", and "Health Behavior" key proteins still remained significant mediating effects (Supplementary Table 14). (lines 189-193)

References:

1. Marassi, M., Fadini, G.P. The cardio-renal-metabolic connection: a review of the evidence. *Cardiovasc Diabetol* 22, 195 (2023).
2. Zhang, R., Mamza, J.B., et al. Lifetime risk of cardiovascular-renal disease in type 2 diabetes: a population-based study in 473,399 individuals. *BMC Med* 20, 63 (2022).
3. Ndumele CE, Neeland IJ, et al. A Synopsis of the Evidence for the Science and Clinical Management of Cardiovascular-Kidney-Metabolic (CKM) Syndrome: A Scientific Statement From the American Heart Association. *Circulation*. 2023 Nov 14;148(20):1636-1664.

Figure R2 Differential Protein Weight Patterns Between Union-Selected and InterSection-Filtered Signatures

Figure R3 Biological Pathway Enrichment in the Union-derived Protein Signatures

- The authors stated that ‘P-value was adjusted by Bonferroni method.’ Please clarify how this adjustment was applied—for example, was it performed per exposure pattern-outcome pair or across all comparisons?

Response:

Thank you for the comment. The correction was applied to all multiple comparisons, adjusting the p-values per exposure pattern-outcome pair to control the overall Type I error rate below 0.05. We have revised the manuscript to express clearly.

- The mediation analysis and the construction of the relationship between exposure patterns and proteins require careful consideration. The authors aim for causal inference, as stated in the Methods: ‘The regression-based causal mediation analysis was carried out to fit the mediation model and the outcome model respectively.’ While the use of time-to-event analysis to establish causal relationships between exposure patterns and outcomes, and between protein scores and outcomes, is appropriate, the use of LASSO regression for exposure pattern-protein relationships is cross-Sectional and cannot infer causality. Additionally, the description of how indirect and direct effects were calculated is unclear. If the indirect effect was calculated by multiplying the LASSO estimate with the Cox regression estimate, this approach may present methodological challenges. I encourage the authors to address these points in detail.

Response:

We sincerely apologize for any confusion caused to the reviewers. As used in high-dimensional mediation analysis^{1, 2}, the LASSO method was used to screen proteins potentially associated with environmental exposures. After the protein selection procedure, we assessed the mediation role of each selected protein in the associations between environmental exposures and health outcomes.

Regarding the cross-Sectional exposure pattern-protein relationships in mediation analysis: The co-exposure patterns extracted in this study aim to reflect the relatively stable and

normative exposure environments. Additionally, some exposure variables (eg. diet, physical activity, and social isolation) were obtained from baseline questionnaires, with questions focusing on "past" and "usual" states, which provides some degree of support for the temporal sequence required for causal inference. We appreciate the reviewers' remarks on this aspect and have incorporated a more detailed explanation in the revised **Limitation Section** (at lines 336 – 341), as follows.

First, the cross-Sectional data limits causal inference. Although the exposure pattern constructed in this study aimed to capture relatively stable, normative environment over time, the assumption of invariance over time against such a long period of time needs to be approached with caution as the questionnaire information was only collected at baseline.

Regarding the direct and indirect effect statistical methods: The estimation of indirect effects was based on the regression-based approach proposed by Valeri et al. (2013) and VanderWeele et al. (2014). Specifically, the estimation of mediation effects involved constructing the following two models: (1) a linear regression model for the relationship between exposure patterns and proteins; and (2) a Cox proportional hazards model for the relationship between exposure patterns, proteins and CMDs risk. Each mediation model examined the relationship between one exposure pattern, one selected protein, and one disease outcome. Therefore, the indirect effect was calculated as an exponent of the product of the linear regression coefficient and the Cox coefficient. We have revised the **Method Section** with a clearer presentation.

In addition, we applied the mediation analysis of the each protein selected by LASSO regression, adjusting for sex, age, ethnicity, alcohol consumption status, and BMI(CMAverse R package). Specifically, the mediation model is a linear model based on the relationship between one co-exposure pattern and one protein, and the outcome model is a Cox proportional hazards model of an exposure pattern, a protein, and a disease. We estimated the natural total indirect effect and mediated proportion through the regression-based approach, and the P-values were adjusted by Bonferroni method within each exposure pattern-outcome pair.(Lines 519-527)

References:

1. Zhu K, Li R, et al. Proteomic signatures of healthy dietary patterns are associated with lower risks of major chronic diseases and mortality. *Nat Food*. 2025 Jan;6(1):47-57.
2. Chen L, Zhernakova DV, et al. Influence of the microbiome, diet and genetics on inter-individual variation in the human plasma metabolome. *Nat Med*. 2022 Nov;28(11):2333-2343.
6. In the mediation analysis, the proportion mediated by several proteins exceeds 100%, with the direct effect being larger than the total effect. This finding warrants careful investigation and interpretation, as it may indicate potential issues with the mediation model.

We appreciate the reviewers' suggestions. The primary reason for this phenomenon is inconsistent mediation (Supplementary Table 7). Specifically, the proportion mediated makes

sense when the sign of the indirect effects is the same as the sign of the direct effects. It is problematic when the sign of the indirect effects and direct effects operate in different directions, which can result in a proportion mediated larger than 100% or even negative and such measure may be not meaningful¹. This situation is also explained in the reference of CMAverse package². We would also like to clarify that all the conclusions of the mediation analysis in our study were drawn after excluding cases where the proportion mediated exceeded 100% or was negative.

To address this ambiguity, we have revised the Result Section. We have also added explanatory notes in the relevant figure (Figure 5) to highlight the instability of the mediation share in these instances. These changes aim to improve the clarity and transparency of our results.

Method :

In addition, we applied the mediation analysis of the each protein selected by LASSO regression, adjusting for sex, age, ethnicity, alcohol consumption status, and BMI(CMAverse R package). Specifically, the mediation model is a linear model based on the relationship between one co-exposure pattern and one protein, and the outcome model is a Cox proportional hazards model of an exposure pattern, a protein, and a disease. We estimated the natural total indirect effect and mediated proportion through the regression-based approach, and the P-values were adjusted by Bonferroni method within each exposure pattern-outcome pair.(Lines 519-527)

Result:

In Fig. 5 and Supplementary Table 8, the feature proteins with substantial mediating effects are presented (adjusted $P < 0.05$ and proportion mediated stable), revealing key mediators between each exposure characteristic and CMDs. (Lines 160-163)

References:

1. VanderWeele, Tyler. Explanation in Causal Inference. Oxford: Oxford University Press, 2015.
 2. VanderWeele, Tyler , Vansteelandt, Stijn. Mediation Analysis with Multiple Mediators. Epidemiologic Methods, vol. 2, no. 1, 2014, pp. 95-115.
7. Finally, while it may be beyond the scope of this manuscript, it is worth noting that leveraging publicly available data—such as epigenome data for environmental exposures, cis-expression quantitative trait loci (cisQTLs) for proteins, and genome-wide association studies (GWAS) for CMDs—to conduct Mendelian randomization (MR) analyses could provide more robust evidence for causal inference.

Response:

We appreciate the reviewers' suggestions, and have added the Mendelian randomization (MR) analysis. Our results support a protective causal effect for the "Blue and Green Space" and "Health Behaviors" patterns. However, the majority of MR analyses examining the

associations between "Air and Noise Pollution", "Social Deprivation", and CMDs did not reach statistical significance, likely for two reasons. First, despite filtering instrumental variables using the F-statistic (threshold = 10), weak instrument bias may still persist^{1,2}. Second, environmental exposures like air pollution, largely determined by geographic factors, are unlikely to be correlated with biological processes influenced by genetic variation, making it challenging to satisfy the assumption of gene-environment equivalence^{3,4}.

The existence of many previous Mendelian randomization studies on the effects of air pollution, educational attainment, and economic income on diseases could, to some extent, support our view of the causality of the co-exposure effect. It is further planned to undertake a more thorough investigation of the effect of co-exposures on health outcomes in future studies. Accordingly, we have revised the relevant sections.

Result

In Supplementary table 3, the Mendelian Randomization analysis revealed that "Health Behaviors" and "Blue and Green Spaces" exhibited protective effects against CMDs, such that "Health Behaviors" were associated with diabetes risk negatively (OR = 0.93 (0.86-1.00)). (Lines 126-130)

References:

1. Skrivankova VW, Richmond RC, et al. Strengthening the reporting of observational studies in epidemiology using mendelian randomisation (STROBE-MR): explanation and elaboration. *BMJ*. 2021 Oct 26;375:n2233.
2. Burgess S, Thompson SG; CRP CHD Genetics Collaboration. Avoiding bias from weak instruments in Mendelian randomization studies. *Int J Epidemiol*. 2011 Jun;40(3):755-64.
3. Lovegrove CE, Howles SA, Furniss D, Holmes MV. Causal inference in health and disease: a review of the principles and applications of Mendelian randomization. *J Bone Miner Res*. 2024 Oct 29;39(11):1539-1552.
4. Davey Smith G. Epigenesis for epidemiologists: does evo-devo have implications for population health research and practice? *Int J Epidemiol*. 2012 Feb;41(1):236-47.

Reviewer #2:

General comment:

Interesting MS. I have a number of comments, however, that may help to improve the work. Especially the biological and pathophysiological relevance of the identified biomarkers should be better explained and presented. Also the major underlying concept, the exposome, is even not mentioned here and many landmark publications are not cited and discussed – a clear weakness of the manuscript.

Response:

We sincerely appreciate the reviewer's insightful comments and suggestions. We have strengthened the discussion on biomarker relevance, incorporated the exposome framework, and cited key landmark studies to enhance the manuscript. Please find details in the

point-by-point response below. **Please note that the line numbers mentioned in the following responses correspond to those in the revised manuscript.**

Specific comments

1. The exposome concept provides the fundamental basis for the present studies. The authors should explain the exposome concept in detail and also cite respective landmark papers on this concept (e.g. PMID: 16103423, PMID: 23519765, PMID: 24659601, PMID: 37165157).

Response:

Thank you for your insightful suggestions. We fully agree with you that the concept of exposome provides an important theoretical basis for this study. In order to better clarify this concept, we have **added citations** and an explanation of the "exposome" in the **Introduction section**.

Introduction:

In real-world co-exposure environments, multiple exposures often interact synergistically or cumulatively, modifying their health impacts when considered alone or in combination^{11, 12, 13, 14, 15}. This has led to the emergence of the study of "exposome", a concept that covers cumulative exposures to physical, chemical, biological and psychosocial factors over a lifetime^{16, 17}. (Lines 31-36)

Plasma proteins, in addition to serving as therapeutic and predictive disease biomolecules^{20, 21}, can provide insights into the intrinsic mechanisms associated with environmental exposures to human health conditions and cardiometabolic diseases^{22, 23}. According to the findings of previous studies, which explored its function at the molecular level in the pathogenesis of various environmental exposures²⁴, the question remains unanswered whether it can similarly reflect the physiological responses to co-exposure patterns and whether it can reveal key biological pathways and mechanisms of the effect on CMDs. (Lines 47-55)

2. Landmark studies on multi-exposure scenarios (PM, UFP, NOx, noise, green space) with respect to cardiometabolic risk should be discussed in detail and respective references must be cited (PMID: 36334460), also published for stroke (PMID: 37265507) and MI (PMID: 37738461) - also graphically summarized in PMID: 37165157.

Response:

We appreciate the reviewer's valuable suggestions. We agree that a more detailed discussion on the impact of multiple exposures on cardiometabolic risk is necessary. Based on the references provided, we have made the following revisions to the manuscript:

In the Introduction Section, we have **additionally cited** PMID: 36334460, PMID: 37265507, and PMID: 37738461 to illustrate the need for comprehensive analyses of multiple environmental exposures.

In the **Discussion Section**, we have expanded on the findings and significance of multiple exposure scenarios.

In multiple exposure scenarios, the health effects of a individual environmental factor are influenced by other factors. The Danish study on stroke and myocardial infarction and the Greece study on cardiometabolic mortality showed effects of green spaces that varied with adjustment for PM_{2.5} or noise^{12,15,14}. (Lines 216-220)

3. Previously, the tight interconnection of different environmental and lifestyle exposures was reviewed in detail, also based on examples for co-localization of major exposure risk factors in the USA (PMID: 34005032).

Response:

Thank you for your comments. The study you mentioned highlights that the co-localization of multiple environmental exposures (e.g., air pollution, noise pollution, and light pollution) may have additive or synergistic effects on cardiovascular health, increasing the risk of chronic non-communicable diseases. As you suggested, this paper provides us with excellent evidence, and we have **added citations** and revised the **Introduction Section** accordingly.

Introduction:

Moreover, widespread exposures usually exhibit a notable colocalization. For example, the distribution of light pollution, air pollution, and traffic noise levels exhibit significant geographical overlap¹⁸. Furthermore, individuals with low socioeconomic status are more prone to engage in detrimental lifestyle behaviors¹⁹. Therefore, given the broad range of exposures, capturing multidimensional co-exposure patterns is essential for understanding the complex relationships between environmental exposures and cardiometabolic health. . (Lines 36 - 43)

4. It would be important if the authors can associate their top candidates of modulated proteins by the different environmental risk factors (e.g. CDCP1, CXCL17, FGF21, GDF15 and IGFBP4 for "Air and Noise Pollution" and "Social Deprivation"; for "blue and green spaces" and "health behaviors" ITGAV, HPGDS, ADGRG2 and MMP12 for the former and CXCL13, CA6, SDC1 and PTN for the latter) with some previous published reports – either animal or human – where these markers were also found modulated by specific exposures.

Response:

Thanks for the comment. In response to the reviewers' suggestions, we conducted a literature review and identified supporting evidence for the candidate proteins. Below is our supplementary explanation:

"Air and Noise Pollution" and "Social Deprivation"

- CDCP1: CDCP1 has been found to be associated with depression, suggesting its role in mental health and environmental stress¹.

- CXCL17: Studies indicate that air pollution can significantly influence CXCL17 expression, highlighting its potential role in inflammation related to environmental pollution².
- FGF21: An association has been observed between FGF21 and PM2.5 exposure, suggesting its potential role in metabolic regulation linked to air pollution³.
- GDF15: As an immune biomarker, GDF15 has been associated with air pollution exposure and blood pressure changes in adolescents. Additionally, short-term exposure to ambient air pollution and fluctuations in outdoor temperature significantly impact myocardial injury, inflammation, and oxidative stress biomarkers (including GDF15) in healthy adults^{4,5}.
- IGFBP4: IGFBP4 is linked to smoking behavior, indicating its potential role in environmental exposure and health behaviors⁶.

"Health Behaviors"

- CXCL13: Lower levels of CXCL13 are associated with higher levels of physical activity^{7,8}.
- CA6: CA6 is associated with salty taste perception and the consumption of sweets and beverages, indicating its role in dietary behavior and metabolic regulation. Further research has examined the biological significance of CA6 in taste perception and eating habits^{9,10}.
- SDC1: SDC1 expression is upregulated with increased exercise intensity, highlighting its role in exercise-related endothelial integrity and activation. Studies have shown that moderate- and high-intensity exercise significantly impact circulating endothelial markers in young, healthy men¹¹.
- PTN: PTN is associated with high-fat diets, suggesting its role in regulating adipocyte precursor activity related to dietary habits¹².

Based on the above content, we have revised the corresponding section in the **Discussion**. Additionally, we have expanded the discussion on proteins related to “Blue and green spaces”.

However, the interpretation of protein in the "Blue and Green Spaces" pattern requires caution. On the one hand, in renal diseases, the protein signature scores inversely correlated with the pattern, suggesting additional explanatory mechanisms beyond these proteins. On the other hand, after adjusting for renal function, only MMP12 retained significant mediation effects. This protein may exert protective effects through immunomodulation and anti-inflammatory pathways, as evidenced by its role in macrophage regulation and extracellular matrix remodeling⁵⁴.(Lines 298-306)

References:

1. Tsang RSM, Timpson NJ, Khandaker GM. Inflammation proteomic profiling of psychosis in young adults: Findings from the ALSPAC birth cohort. *Psychoneuroendocrinology*. 2025 Jan;171:107188.
 2. Yin Y, Mu C, et al. CXCL17 Attenuates Diesel Exhaust Emissions Exposure-Induced Lung Damage by Regulating Macrophage Function. *Toxics*. 2023 Jul 26;11(8):646.
 3. Divorcy N, Mackenzie AE, Nicklin SA, Milligan G. G protein-coupled receptor 35: an emerging target in inflammatory and cardiovascular disease. *Front Pharmacol*. 2015 Mar 10;6:41.
 4. Prunicki M, Cauwenberghs N, et al. Immune biomarkers link air pollution exposure to blood pressure in adolescents. *Environ Health*. 2020 Oct 16;19(1):108.
 5. Xu H, Brook RD, et al. Short-term effects of ambient air pollution and outdoor temperature on biomarkers of myocardial damage, inflammation and oxidative stress in healthy adults. *Environ Epidemiol*. 2019 Dec 2;3(6):e078.
 6. Chelchowska M, Maciejewski T, et al. The pregnancy-associated plasma protein A and insulin-like growth factor system in response to cigarette smoking. *J Matern Fetal Neonatal Med*. 2012 Nov;25(11):2377-80.
 7. Saberi Hosnijeh F, Koliijn PM, et al. Mediating effect of soluble B-cell activation immune markers on the association between anthropometric and lifestyle factors and lymphoma development.
 8. Wang SS, Zhong C, et al. Host characteristics associated with serologic inflammatory biomarkers in women. *Cytokine*. 2022 Jan;149:155726.
 9. Morzel M, Truntzer C, et al. Associations between food consumption patterns and saliva composition: Specificities of eating difficulties children. *Physiol Behav*. 2017 May 1;173:116-123.
 10. Méjean C, Morzel M, et al. Salivary Composition Is Associated with Liking and Usual Nutrient Intake. *PLoS One*. 2015 Sep 4;10(9):e0137473.
 11. Sapp RM, Evans WS, et al. The effects of moderate and high-intensity exercise on circulating markers of endothelial integrity and activation in young, healthy men. *J Appl Physiol (1985)*. 2019 Nov 1;127(5):1245-1256.
 12. Wong JC, Krueger KC, et al. A glucocorticoid- and diet-responsive pathway toggles adipocyte precursor cell activity in vivo. *Sci Signal*. 2016 Oct 25;9(451):ra103.
5. Figure 2: When looking at the hazard ratios for the different environmental & lifestyle risk factors a certain discrepancy between the present findings and the reports by the GBD study become evident. In the GBD air pollution is meanwhile a leading driver of global deaths and burden of disease – just at the same level as tobacco smoking. This is not reflected at all by the present data. How can this be explained?

Response:

We sincerely appreciate the reviewer's insightful comments. Our findings do not contradict the GBD.

First, our estimates regarding the harmful effects of smoking and air pollution remain

consistent with previous studies, suggesting that smoking may present a greater health risk than air pollution on etiology. For example, studies on the health hazards of smoking in China, Europe, and the United States reported hazard estimates of 1.58, 1.6, and 3.0, respectively¹⁻³, which are very close to our findings. Similarly, studies have shown that for every 10 µg/m³ increase in PM_{2.5}, the HR for type 2 diabetes is 1.15, and for chronic kidney disease is 1.08⁴. Even in studies assessing both risk factors simultaneously, the adjusted relative risk (95% CI) for cardiovascular disease associated with smoking (≤ 3 [1.5] cigarettes/day vs. no smoking) was 1.64 (1.42–1.89), exceeding the corresponding risk for air pollution (1.12 [1.08–1.15] per 10 µg/m³ increase)⁵.

Second, the disease burden estimation (which GBD indicated) fundamentally follows a double-factor framework: 1) etiological effect magnitude (quantified by HR/RR/OR values), and 2) population exposure distribution patterns (reflected through incidence/prevalence differentials). The comparison between air pollution and smoking illustrates this framework. While air pollution demonstrates a lower etiological effect magnitude, as indicated by smaller HR values, its extensive population exposure results in disease burdens comparable to smoking⁵.

This study specifically focuses on elucidating etiological associations between individual risk factors and disease outcomes (HR quantification) and thus is reasonably inconsistent with the GBD findings. Evaluating the importance of each exposure factor from a public health perspective also requires a comprehensive consideration of the prevalence of each exposure factor in the population.

References:

1. Zhang L, Ma Y, et al. Tobacco smoke and all-cause mortality and premature death in China: a cohort study. *BMC Public Health*. 2023 Dec 12;23(1):2486.
 2. Argentieri MA, Amin N, et al. Integrating the environmental and genetic architectures of aging and mortality. *Nat Med*. 2025 Feb 19.
 3. Rajagopalan S, Brook RD, et al. Air pollution exposure and cardiometabolic risk. *Lancet Diabetes Endocrinol*. 2024 Mar;12(3):196-208.
 4. Carter BD, Abnet CC, et al. Smoking and mortality--beyond established causes. *N Engl J Med*. 2015 Feb 12;372(7):631-40.
 5. Pope CA 3rd, Burnett RT, et al. Cardiovascular mortality and exposure to airborne fine particulate matter and cigarette smoke: shape of the exposure-response relationship. *Circulation*. 2009 Sep 15;120(11):941-8.
 6. GBD 2017 Risk Factor Collaborators. Global, regional, and national comparative risk assessment of 84 behavioural, environmental and occupational, and metabolic risks or clusters of risks for 195 countries and territories, 1990-2017: a systematic analysis for the Global Burden of Disease Study 2017. *Lancet*. 2018 Nov 10;392(10159):1923-1994.
6. Since these are data from the UK Biobank I assume that recruitment was done in the UK? Please provide a map providing information on the regional distribution of the recruited study population (can be provided in the supplement).

Response:

Thank you for your suggestion. The UK Biobank recruited participants from across the UK, with assessments conducted in 22 centres in Scotland, England, and Wales at baseline. The distribution of the sample site was shown as follows and has been added in the **Supplementary Method**.

Figure 1 for Supplementary Methods

Due to data licensing restrictions related to participants' home locations, we are unable to provide a map showing the regional distribution of participants. However, we have reviewed the literature and present maps from other studies below. The UK Biobank initially planned for 35 assessment centers across the UK (Figure R4 A), but ultimately established 22 centers during the baseline assessments conducted from 2006 to 2010 (Figure R4 B)¹. Additionally, researchers² created a density heat map of the UK Biobank participants, complete with grid coordinates (Figure R4 C).

Figure R4 Maps of assessment centers and population density^{1,2}

References:

1. UK Biobank: Protocol for a large-scale prospective epidemiological resource (Protocol No: UKBB-PROT-09-06). UK Biobank Coordinating Centre. 2006.

2. Wong JYY, Jones RR, et al. Commute patterns, residential traffic-related air pollution, and lung cancer risk in the prospective UK Biobank cohort study. *Environ Int.* 2021 Oct;155:106698.
7. Exposure maps must be provided for the following environmental risk factors (can be presented in the supplement): NO_x, NO₂, PM₁₀, PM_{2.5}, Noise, green space, blue space.

Response:

Thank you for your suggestion. We have included box plots of various pollution indicators (NO_x, NO₂, PM₁₀, PM_{2.5}, noise, green space, blue space) in the **Supplementary Method** to illustrate the distribution of exposures using the available data.

Figure 2 for Supplementary Methods

Due to data access restrictions from the UKB, we are unable to directly provide exposure maps based on geographical location. Instead, we presented maps from other studies in Figure R5, hoping that will help you better understand the spatial distribution of air pollution exposure in the UKB study.

[REDACTED]

.Figure R5 Maps of air pollution ¹

References:

1. Zhang J, Fang XY, et al. Association of Combined Exposure to Ambient Air Pollutants, Genetic Risk, and Incident Rheumatoid Arthritis: A Prospective Cohort Study in the UK Biobank. *Environ Health Perspect*. 2023 Mar;131(3):37008.
8. A summarizing scheme would be helpful to highlight the major (patho)physiological pathways triggered by the different top candidates for the specific exposures. This would allow the reader to better understand the individual pathomechanisms triggered by the different exposures.

Response:

Thank you for your suggestion. We added a summarizing diagram to the **Supplementary Material** focusing on the key physiological and pathological pathways triggered by different major exposure candidates.

Supplementary Fig. 1 Exposure Patterns Corresponding to the Top Five Proteins with the Highest Mediated Proportions for Each Disease

Dear Reviewers,

Thank you very much for the opportunity to revise our manuscript. We are grateful for the constructive and insightful comments provided by the reviewers. We have carefully addressed all the reviewers' suggestions and revised the manuscript accordingly.

In this response letter, our point-by-point replies to the reviewers' comments are provided in **blue regular font**. Text quoted from the revised manuscript is shown in *blue italic font*, with corresponding line numbers indicated in parentheses. In the revised manuscript, all modifications have been marked in **red** for clarity.

We hope that the revised version meets the expectations of both the reviewers. Thank you for your time and consideration.

REVIEWER COMMENTS

Reviewer #1:

The authors have conducted a thorough revision to response my comments, which is greatly appreciated. The paper now has been substantially improved.

Response:

We sincerely thank the reviewer for the positive feedback and constructive suggestions, which have greatly helped improve the quality of our manuscript.

I agree with the authors' response regarding the inappropriateness of using GWAS of environmental exposures for Mendelian randomization analysis, and that was why I mentioned the 'epigenome data for environmental exposures' instead of 'GWAS data of environmental exposures'. Even though the MR results were under expectation, I suggest to remove the MR part of the study but simply mention it in the discussion as future investigations.

Response:

We appreciate the reviewer's clarification and constructive suggestion. Following the recommendation, we have removed the MR analysis from the main text and instead mentioned it in the discussion section as a potential direction for future studies. We have made corresponding revisions in the Discussion Section:

Finally, future research could leverage available data—such as epigenomic markers as environmental exposure proxies, protein cis-QTLs, and cardiometabolic GWAS—in Mendelian randomization analyses to strengthen causal validation of environment-protein-disease pathways.(Line 359 - 363)

Reviewer #2 :No further comments

Response:

We appreciate the reviewer's time and previous feedback, and we are grateful for their continued consideration of our work.

Reviewer #3 :

Thanks to the author team for their work. This is indeed an important piece of work, and I think the authors have addressed all reviewers' comments. I'm happy with those modifications.

Response:

We greatly appreciate the reviewer's kind words and thoughtful evaluation of our work.

However, a few minor comments from me:

1. In lines 446–448, please refer to existing literature to support the method you use to impute the missing data.

Response:

Thank you for your valuable feedback. In response to your comment, we have now incorporated citations to relevant literature supporting the imputation method used for handling missing values in the proteomic data. This addition appears in Line 452 - 454 of the revised manuscript.

2. In the Cox proportional hazards regression model from your “Effects and contributions of individual exposures” section, you adjusted for BMI and age as continuous variables. That is based on the hypothesis that the effects of age and BMI are linear. Please consider using a non-linear method to adjust these confounders—e.g., polynomial terms or categorical variables.

Response:

We appreciate the reviewer's valid concern regarding potential nonlinear confounding effects of age and BMI. To address this, we conducted sensitivity analyses using categorized versions of these variables: age was classified as “younger” (<60 years) and “older” (≥60 years), and BMI was grouped into four categories: underweight (<18.5 kg/m²), normal (18.5–24.9 kg/m²), overweight (25.0–29.9 kg/m²), and obesity (≥30 kg/m²). The resulting hazard ratios were consistent with those from the primary models, with all estimates varying by less than ±5%, demonstrating consistency across different specifications of age and BMI. These results have been added to the sensitivity analysis section and are presented in Supplementary Table 8.

First, to examine the assumption of linearity for continuous confounders, we conducted Cox proportional hazards models using categorized age and BMI. Specifically, age was categorized as <60 and ≥60 years, and BMI as underweight (<18.5 kg/m²), normal (18.5–24.9 kg/m²), overweight (25.0–29.9 kg/m²), and obesity (≥30 kg/m²). (Line 536 - 541)

REVIEWERS' COMMENTS

Reviewer #3 (Remarks to the Author):

I'm happy with those modifications. No further comments